# Developmental seizures and mortality result from reducing GABA$_A$ receptor α2-subunit interaction with collybistin

Rochelle M. Hines [1,2], Hans Michael Maric [3,4], Dustin J. Hines[1,2], Amit Modgil[1], Patrizia Panzanelli[5], Yasuko Nakamura[1], Anna J. Nathanson[1], Alan Cross[6], Tarek Deeb[1,7], Nicholas J. Brandon[6,7], Paul Davies[1], Jean-Marc Fritschy [8,9], Hermann Schindelin [3] & Stephen J. Moss[1,7,10]

Fast inhibitory synaptic transmission is mediated by γ-aminobutyric acid type A receptors (GABA$_A$Rs) that are enriched at functionally diverse synapses via mechanisms that remain unclear. Using isothermal titration calorimetry and complementary methods we demonstrate an exclusive low micromolar binding of collybistin to the α2-subunit of GABA$_A$Rs. To explore the biological relevance of collybistin-α2-subunit selectivity, we generate mice with a mutation in the α2-subunit-collybistin binding region (*Gabra2*-1). The mutation results in loss of a distinct subset of inhibitory synapses and decreased amplitude of inhibitory synaptic currents. *Gabra2*–1 mice have a striking phenotype characterized by increased susceptibility to seizures and early mortality. Surviving *Gabra2*-1 mice show anxiety and elevations in electroencephalogram $\delta$ power, which are ameliorated by treatment with the α2/α3-selective positive modulator, AZD7325. Taken together, our results demonstrate an α2-subunit selective binding of collybistin, which plays a key role in patterned brain activity, particularly during development.

[1] Department of Neuroscience, Tufts University School of Medicine, Boston 02111 MA, USA. [2] Department of Psychology, University of Nevada Las Vegas, Las Vegas 89154 Ne, USA. [3] Rudolf Virchow Center for Experimental Biomedicine, University of Würzburg, Würzburg D-97080, Germany. [4] Department of Biotechnology and Biophysics, Biocenter, University of Würzburg, Würzburg D-97080, Germany. [5] Department of Neuroscience Rita Levi Montalcini, University of Turin, Turin 10126, Italy. [6] AstraZeneca Neuroscience iMED, Biotech Unit, Boston 02451 MA, USA. [7] AstraZeneca Tufts Laboratory for Basic and Translational Neuroscience, Boston 02111 MA, USA. [8] Institute of Pharmacology and Toxicology, University of Zurich, Zurich 8057, Switzerland. [9] Center for Neuroscience Zurich, University of Zurich and ETH Zurich, Zurich 8057, Switzerland. [10] Department of Neuroscience, Physiology and Pharmacology, University College, London WC1E 6BT, UK. Correspondence and requests for materials should be addressed to R.M.H. (email: Rochelle.Hines@unlv.edu) or to S.J.M. (email: Stephen.Moss@tufts.edu)

The brain functions via a dynamic partnership between excitatory principal cells and inhibitory interneurons, ensuring regulation of global cell firing rates, yet allowing for local control of excitability. Fast inhibitory synaptic transmission is mediated by γ-aminobutyric acid type A receptors (GABA$_A$Rs), which are composed of combinations of subunit families α(1–6), β(1–3), γ(1–3), δ, ε, θ, and π[1]. Despite GABA$_A$R-α-subunits being highly homologous, they are known to be specifically localized, with GABA$_A$Rs containing α1 enriched at dendritic and somatic synapses, and α2 localized to synapses on the axon initial segment (AIS)[2–4]. It has been long speculated that α-subunits are prime mediators in selective targeting[5]; however, the precise mechanisms regulating this targeting remain unclear. Comparison of α1 and α2 structures reveals that the large intracellular loop between transmembrane domains 3 and 4 is the most divergent, but its necessity for targeting is unclear. Given the important role of AIS synapses in regulating excitatory cell output[6,7], and that α-subunit isoforms determine the decay of inhibitory postsynaptic currents[8], enrichment of α2 to AIS synapses may profoundly affect neuronal excitability. Therefore, it is of fundamental importance to understand how neurons regulate the formation of synapse subtypes and control selective targeting of α2-containing GABA$_A$Rs.

A well-known inhibitory synaptic organizer, gephyrin, has been shown to interact with multiple α-subunits of GABA$_A$Rs[9–12]. Structural and mutational studies have identified a universal binding site within gephyrin that allows for the alternative recruitment of glycine and GABA$_A$R subunits[9]. Mutations that weaken the receptor–gephyrin interaction have been shown to alter receptor accumulation, diffusion, clustering, and transport, and can be related to severe phenotypes, including epilepsy[13–15]. Early studies demonstrated a link between the GABA$_A$R-α2-subunit and gephyrin[11], however, quantification of the interaction later revealed a very low interaction strength[16]. Collybistin (CB; ARHGEF9) has also been shown to interact with gephyrin, as well as GABA$_A$Rs[17–20], but quantification of the interaction strength has not been possible. CB and its human homolog hPEM-2 (human homolog of posterior end mark-2) are guanine nucleotide exchange factors that activate the small GTPase CDC-42, regulating the actin cytoskeleton[21]. CB is generally composed of a catalytic Rho guanine nucleotide exchange factor domain, a pleckstrin homology (PH) phosphoinositide-binding domain, as well as an Src homology 3 (SH3) domain. Yeast-two-hybrid (Y2H) assays suggested the GABA$_A$R-α2-subunit[19] and neuroligin-2[22] as possible activators for CB by directly interacting with the SH3 domain, resulting in its release from the PH and DH domains. Mapping studies proposed an overlapping binding site for gephyrin and CB on the large intracellular loop of the GABA$_A$R-α2-subunit, and it has been suggested that synergistic binding could produce a tripartite complex[19].

The study of both mouse models and human neurodevelopmental syndromes characterized by CB loss supports a functional link to GABA$_A$Rs. Deletion of Arhgef9 in mice leads to selective reductions in the number of inhibitory synapses within the hippocampus[21,23]. A naturally occurring missense mutation of ARHGEF9 (G55A) is associated with hyperekplexia, epilepsy, and mental retardation[24,25], and has been shown to disrupt the clustering of α2-containing GABA$_A$Rs and gephyrin[19]. Similarly, a translocation mutation within the ARHGEF9 gene has been shown to result in a neurodevelopmental syndrome that includes epilepsy, anxiety, aggressive behavior, sleep–wake disruptions, and mental retardation, but not hyperekplexia[24,26,27], suggesting a broad disruption of GABAergic but not glycinergic signaling in this variant of the disorder. Despite these findings, the respective affinity of CB for specific GABA$_A$R subunits remains unknown[24,28]. Further, the precise molecular and cellular mechanisms mediating the severe symptoms seen following mutations in ARHGEF9 remain unknown.

Using isothermal titration calorimetry and complementary methods, we identify an exclusive binding of CB to the GABA$_A$R-α2-subunit. Notably, this is the first report of a high-affinity protein interaction for both CB and α2. To reveal the biological relevance, we create a novel knock-in mouse where amino acids 358–375 from the α1 loop are substituted into α2 (Gabra2-1), a substitution that reduces CB binding. This mutation alters the expression of α2 and CB, inhibitory current amplitude and decay time, and results in the loss of AIS synapses. Gabra2-1 mice display a remarkable phenotype characterized by spontaneous seizures and early mortality, reminiscent of the human syndrome linked to ARHGEF9 mutation. Surviving Gabra2-1 mice do not show behavioral seizures, but are more susceptible to seizure induction by kainate. Gabra2-1 mice also display anxiety-like behavior, and an enhancement of power in the δ-band using electroencephalogram (EEG) recording, both of which are ameliorated by treatment with the α2/α3-selective GABA$_A$R positive modulator AZD7325.

## Results

**Collybistin binds the GABA$_A$R-α2-subunit with high affinity**. Quantitative binding studies have shown that gephyrin displays a subunit binding profile that dictates receptor recruitment[9,16]. Prompted by Y2H studies reporting GABA$_A$R-CB interactions[19], we speculated that CB might display subunit specificity. We used purified proteins, including the GABA$_A$R-α1, -α2, and -α3 loops, the SH3 domain of CB (CB-SH3), and the E domain of gephyrin (GephE; Fig. 1a). We applied mixtures of purified CB-SH3 and the intracellular loops of GABA$_A$R-α1–3 to native agarose gel electrophoresis (NAGE) for identification of low affinity interactions by altered migration of the native proteins upon complex formation. While the SH3-GABA$_A$R-α1L and -α2L complexes are retained in the pocket, the GABA$_A$R-α3-subunit allows SH3 to enter the gel (Fig. 1b), suggesting that -α1L and -α2L (but not -α3L) are capable of forming a complex with CB-SH3.

To estimate the stability of the GABA$_A$R-SH3 complexes we further characterized them by analytical size exclusion chromatography (SEC). Either single proteins (Fig. 1c top) or equimolar mixtures of the SH3 domain combined with either -α1L, -α2L, or -α3L were applied to a column and the resulting fractions were analyzed by SDS-PAGE. While the NAGE-assay revealed an SH3-α1L complex, SEC analysis demonstrated its low stability, as the SH3 domain elution profile was unaltered by -α1L (Fig. 1c). In line with the NAGE experiments, -α3L did not alter SH3 elution (Fig. 1c). SEC demonstrated a tight SH3-α2L complex, resulting in a significant 10 ml shift of the SH3 domain toward higher molecular weights and correspondingly lower elution volumes (Fig. 1c). Taken together, the SEC and NAGE results define a weak SH3-α1L complex and a tight SH3-α2L complex.

Encouraged by the NAGE and SEC, we next quantified the interactions using isothermal titration calorimetry (ITC; Fig. 1d). ITC revealed that the -α2L forms a tight complex ($K_d = 1$ μM), while -α1L forms a rather weak complex ($K_d \sim 500$ μM; Table 1). To narrow down the GABA$_A$R-α2 binding site, we determined the SH3 affinity of different truncated constructs. The first half of the receptor large intracellular loop displayed an affinity comparable to the full-length receptor loop (Supplementary Table 1). Further N-terminal truncations of this construct significantly decreased the SH3 affinity (Supplementary Table 2), suggesting an important role of secondary structure and/or cooperative effects for this interaction.

Based on the proposal of a tripartite complex[19], and the overlapping binding site of GephE and CB-SH3 on -α2L, we

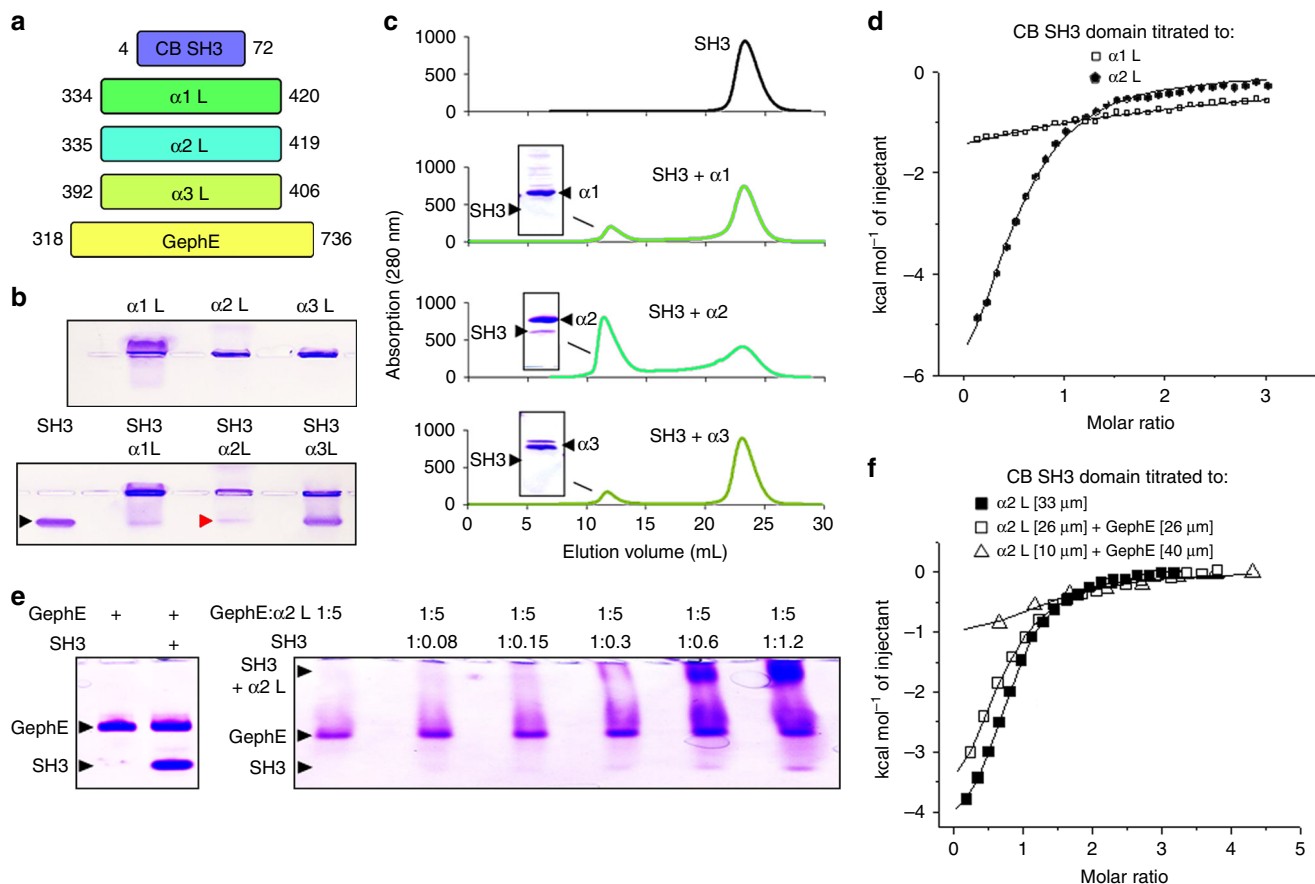

**Fig. 1** Analyzing the interaction of $GABA_A R$ α subunits with collybistin. **a** Cartoon showing the purified proteins used for biochemical studies of α-subunit interaction with gephyrin and CB. **b** Receptor subunits were subject to NAGE (pockets mark sites of protein application) followed by coomassie staining, revealing that α-subunit loops slowly migrate to the cathode. CB-SH3 alone or mixed with a threefold excess of $GABA_A R$ intracellular domains reveals that α2L markedly alters the migration of SH3 (red arrow). **c** SEC analysis was used to explore the behavior of CB-SH3 alone or when mixed with equimolar amounts of α1L, α2L, or α3L. Fusion proteins were detected in eluates using absorbance at 280 nM and coomassie staining after SDS-PAGE (insets), verifying the presence of the SH3 domain at lower elution volumes in the fraction containing α2L. **d** CB-SH3 was titrated against α1L and α2L. The measured binding enthalpies are plotted as a function of the molar ratio of the SH3 domain to the $GABA_A R$ α1 and α2 loops. **e** GephE and GephE + CB-SH3 were subject to NAGE. The migration of the respective proteins is indicated. In presence of the SH3 domain the migration of gephyrin is not altered indicating that the proteins do not interact on NAGE (left panel). GephE in the presence of a fivefold molar excess of α2L is partly retained in the pocket and partly migrates towards the anode (lane 1 right panel). Addition of increasing amounts of the SH3 domain to the GephE-α2L complex allows more gephyrin to enter the gel and migrate towards the anode. At the same time SH3 binding to α2L retains SH3 in the pocket. This indicates that SH3 and gephyrin compete for α2 binding. **f** CB-SH3 was titrated against α2L alone or in combination with GephE, and the measured binding enthalpies are plotted

**Table 1 $K_d$ values for the α1–3 subunits and glycine receptors with gephyrin (GephE) or CB (CB-SH3) are compared**

| Receptor | Binding motif | GephE $K_d$ | CB-SH3 $K_d$ |
|---|---|---|---|
| $GABA_A R$-α1 | LIKKNNTYAPTATSYT | 17.0 ± 11.0 | 500 ± 400 |
| $GABA_A R$-α2 | VMIQNNAYAVAVANYA | >500 | 1.3 ± 0.8 |
| $GABA_A R$-α3 | AKKTSTTFNIVGTTYP | 5.0 ± 2.0 | >500 |
| GlyR | NDLRSNDFSIVGSLPR | 0.1/8.0 | >500 |

Values listed are mean ± standard error

probed these interactions for possible synergistic or competitive binding. This set of experiments revealed that GephE and CB-SH3 do not interact but instead compete for -α2L (Fig. 1e). After verification of an overlapping -α2L binding site for GephE and CB-SH3, we explored whether GephE can modulate the binding of SH3-α2L. Therefore, we titrated -α2L in the presence or absence of GephE with the SH3 domain (Fig. 1f). While the heat signatures were altered by differing concentrations of the -α2L, curve fitting revealed that the interaction strength remained unaltered in the presence of gephyrin (Fig. 1f). To compare the interaction of the full-length proteins, we performed GFP bead trap assays with pHlorin-tagged-α2, or a mutant form of pHlorin-tagged-α2, wherein amino acids 358–375 within the intracellular loop were replaced with those from $GABA_A R$-α1 (α2–1; Fig. 2), which were expressed along with CB in human embryonic kidney (HEK) cells. We found that pHlorin-α2 could coimmunoprecipitate CB, while pHlorin-α2–1 coimmunoprecipitated significantly lower amounts of CB (Fig. 2b, c). These results validate that amino acids 358–375 within the intracellular loop of $GABA_A R$-α2 are also important for interaction with CB in the context of the full-length proteins. Taken together, we establish a complementary subunit preference of CB compared to gephyrin, and competition between both proteins for α2-subunit binding.

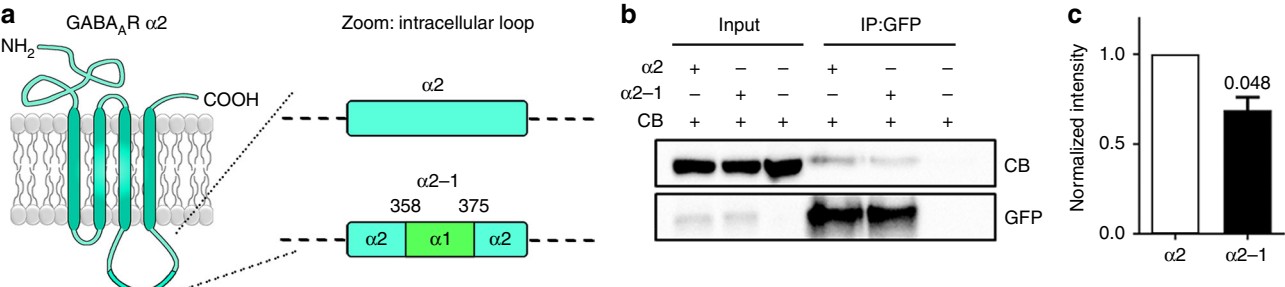

**Fig. 2** Substitution of 13 amino acids from the GABA$_A$R-α1 subunit loop into α2 reduces collybistin binding. **a** Cartoon of the GABA$_A$R α2 subunit, showing the N- and C-termini and the large intracellular loop, expanded to show the site of the α2–1 mutation at amino acids 358–375. **b** Purified GFP-tagged α2 or α2–1 was combined with purified CB and subjected to coimmunoprecipitation. **c** Quantification of normalized intensities comparing GFP-tagged α2 or α2–1 (0.6895 ± 0.08221) interacting with CB. All plots shown and all values listed are mean ± standard error, $t$ test

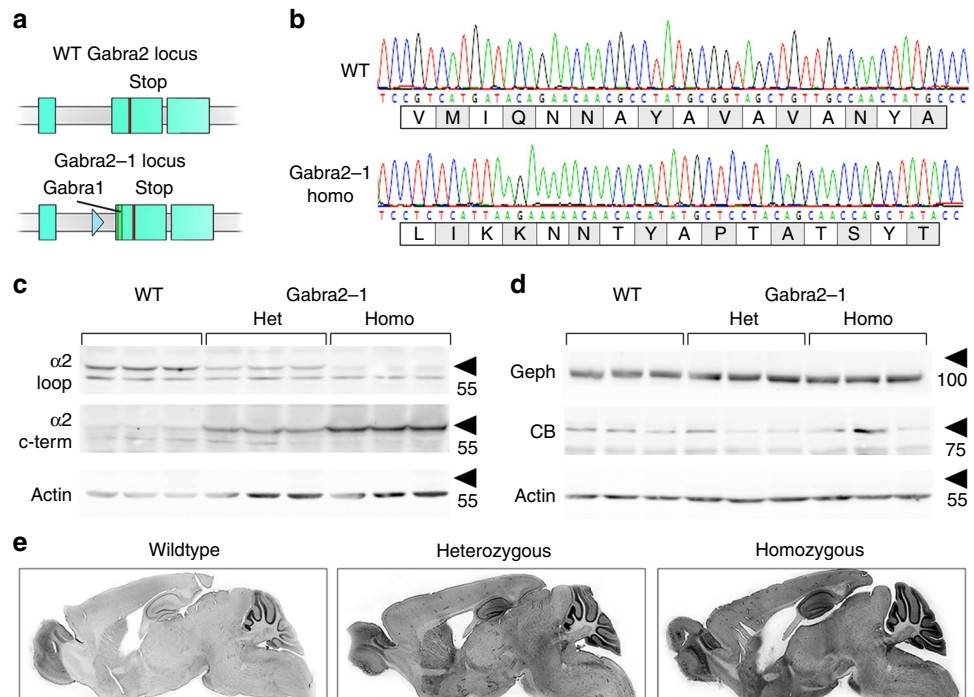

**Fig. 3** Generation and basic characterization of α2–1 knock-in (*Gabra2*–1) mice. **a** Cartoon showing the targeting vector used to insert residues 358–375 from the α1 subunit into exon 10 of the α2 subunit, and the resulting allele. **b** Sequencing of PCR amplified genomic DNA from wildtype and *Gabra2*–1 homozygous mice, and predicted amino acid sequence. **c** Hippocampal extracts from wildtype and *Gabra2*-1 heterozygous and homozygous mice were immunoblotted with antibodies directed at the large intracellular loop (loop) and the c-terminus (c-term) of the GABA$_A$R α2 subunit. **d** Immunoblotting with antibodies for gephyrin (Geph) and collybistin (CB) in wildtype and *Gabra2*-1 mice. See Supplementary Fig. 2 for uncropped blots and quantification of immunoblotting. **e** The 40 μm saggital sections immunostained with α2 c-term antibody and HRP-conjugated secondary antibody, and visualized using 3, 3′-diaminobenzidine staining. Scale bar = 1500 μm. See Supplementary Fig. 3 for quantification of α2 immunohistochemistry

**A mouse model with reduced collybistin-GABA$_A$R-α2 binding.** After having identified a tight complex between CB and the GABA$_A$R-α2-subunit, we next wanted to investigate the functional and physiological relevance of this interaction, due to the implications of *ARHGEF9* mutations in human disease[24,26,27]. To do so, we generated a knock-in mouse where amino acids 358–375 within the large intracellular loop of GABA$_A$R-α2 were replaced by those of GABA$_A$R-α1 (*Gabra2*–1; Fig. 3a).

To confirm the *Gabra2*–1 mutation we amplified this portion of genomic DNA from wildtype and homozygous *Gabra2*–1 mice and sequenced the polymerase chain reaction (PCR) product (Fig. 3b; Supplementary Fig. 1B, C). Sequencing results showed the site of mutation, and the predicted amino acid sequence

alignments revealed insertion of the gephyrin-binding motif (GABA$_A$R-α1 loop) into α2[10] (Fig. 3b), comparable to the alignment of α1 and α2 (Supplementary Fig. 1D). Further validation was obtained using an antibody that recognizes the α2-subunit loop (Fig. 3c, Supplementary Fig. 2), which showed a reduction in antibody recognition by a factor of ~2 in hippocampal lysates from heterozygous mice, and a complete loss in homozygotes. Using an antibody that recognizes the α2-subunit c-terminus we could demonstrate that α2 expression is maintained in *Gabra2*–1 mice (Fig. 3c). We also examined the expression of gephyrin and collybistin in hippocampal lysates and found that although total gephyrin was unchanged, collybistin decreased in expression in *Gabra2*–1 mice relative to control

(Fig. 3d). To examine the regional expression of α2, we conducted DAB immunostaining using a GABA$_A$R-α2-subunit c-terminal antibody comparing *Gabra2*–1 mice to wildtype controls (Fig. 3e, Supplementary Fig. 3A). GABA$_A$R-α2-subunit c-terminal immunoreactivity was generally increased in *Gabra2*–1, with cortex, basal forebrain, and hippocampus showing increases (Supplementary Fig. 3B, C). Within the hippocampus, GABA$_A$R-α2-subunit immunoreactivity was increased along the cell body layer of CA1, and in the molecular layer (Supplementary Fig. 3C). These studies confirmed effective *Gabra2*–1 loop mutation, and demonstrated that the mutated α2-subunit is stably expressed.

**Mislocalization of GABA$_A$R-α2 and collybistin in Gabra2-1.** To investigate clustering of inhibitory synaptic proteins we conducted immunofluorescent staining for GABA$_A$R-α2, -α1, CB, and the vesicular GABA transporter (VGAT; Fig. 4). Consistent with our immunoblotting and DAB immunohistochemistry (IHC) findings, we saw an increase in the total number of α2 clusters (Fig. 4a, c), although these clusters were reduced in size (Fig. 4b). Examination of α1-subunit immunofluorescence revealed no significant change in either the size or density of positive clusters (Fig. 4d, e; Supplementary Fig. 4A, B). The relationship between cluster size and intensity revealed a shift toward an increased number of smaller clusters of α2 in *Gabra2*–1, whereas the α1 relationship appears unaltered (Fig. 4e). Examination of CB immunostaining revealed a dramatic reduction in CB positive cluster size and number (Fig. 4f, h, i), while immunostaining for VGAT revealed no significant change in *Gabra2*–1 (Fig. 4g, j; Supplementary Fig. 4C). As a secondary measure of inhibitory presynaptic terminals we also stained for glutamate decarboxylase (GAD-65), which was also unaffected by the *Gabra2*–1 mutation (Supplementary Fig. 4F). To further evaluate inhibitory synapses we also assessed the colocalization of gephyrin and α2, as well as VGAT and α2. No change was observed in either the percentage of gephyrin colocalized with α2, or VGAT colocalized with α2 (Supplementary Figs. 5 and 6). A decrease in the percentage of α2 clusters colocalized with both gephyrin (Supplementary Fig. 5), and VGAT (Supplementary Fig. 6) declined upon *Gabra2*-1 mutation, which may reflect the increased expression and more diffuse localization of α2. These changes, however, do not reflect a loss of total numbers of inhibitory synapses, evidenced by unchanged VGAT and GAD-65 immunoreactivity, and maintenance of VGAT, and gephyrin, colocalization with α2. Taken together, the replacement of the CB binding site in α2 resulted in loss of CB and altered GABA$_A$R-α2-subunit cluster distribution.

**Smaller sIPSC amplitude and faster decay time in Gabra2-1.** Our immunoblotting and IHC experiments demonstrate modifications in the expression and clustering of α2. Based on these findings we assessed the impact of the *Gabra2*–1 mutation on inhibitory synaptic signaling. To address this, we recorded spontaneous inhibitory synaptic currents (sIPSCs) in CA1 principal neurons of hippocampal slices at postnatal day (PND) 21[29,30] (Fig. 5). Quantification of sIPSC amplitudes revealed a significant decrease in *Gabra2*-1 heterozygous and homozygous slices compared to littermate controls (Fig. 5b, c). Analysis revealed a decrease of sIPSC decay times in CA1 neurons compared to littermate controls (Fig. 5d, e). Previous studies have indicated that the α-subunit in part determines the decay of GABA$_A$R-mediated currents, thus this result may suggest a replacement of α2- with α1-subunits in CA1[8,31]. Assessments of sIPSC frequency did not reveal a significant difference between *Gabra2*–1 slices and littermate controls (Supplementary Fig. 7). These results reveal that despite no loss of inhibitory synaptic

contacts, *Gabra2*–1 mice have compromised inhibitory synaptic transmission that may be related to a decrease in α2 cluster size.

**Loss of GABA$_A$R-α2 clusters at specific types of synapses.** GABA$_A$Rs containing α2 have been identified at a variety of different types of synaptic contacts, but are strongly enriched at synaptic contacts onto the AIS[2–4], hence we examined the subcellular localization of α2. We began by culturing cortical cells to more readily identify the AIS (pan-Na$^+$ channel antibody), and colabeled for the GABA$_A$R-α2 subunit, which revealed a dramatic loss of α2-subunit clusters on the AIS in *Gabra2*–1 derived cultures (Fig. 6a, b). We followed with colocalization of α2 with PV (marker for interneurons including axo-axonic chandelier cells and terminals), and CB1R (marker for CCK positive basket cell terminals) in cortical sections. Overlay images reveal segregated populations of GABA$_A$R-α2-subunit clusters (Fig. 6c) where α2 can be seen colocalized with either PV (teal–green + blue), or CB1R (yellow–green + red) in zoomed panels. Despite a total increase in GABA$_A$R-α2 cluster density in *Gabra2*-1 mice, there was a significant decrease in the percent of α2 colocalized with PV (Fig. 6c, d). Percent colocalization with CB1R positive clusters was not altered (Fig. 6c, e). These findings suggest a specific loss of α2 positive clusters opposing contacts from PV positive cells, including chandelier cells. To follow up on this, we examined the number of inhibitory presynaptic contacts stained with VGAT, onto the AIS of cortical pyramidal cells (Fig. 6f). We found that both heterozygous and homozygous cells showed a significant reduction in the number of VGAT positive clusters per 5 μm AIS (Fig. 6f, g). The number of VGAT positive clusters per 100 μm$^2$ of soma was unaffected (Supplementary Fig. 4D, E). These data indicate a selective loss of α2 opposed to PV positive contacts, paralleled by a reduction of VGAT positive clusters along the AIS.

**Developmental seizure susceptibility and mortality in Gabra2–1.** *Gabra2*–1 mice are viable and fertile, but a subset of heterozygous and homozygous Gabra2–1 pups die prior to weaning, with the peak of mortality falling on PND 20 (Fig. 7a). At PND 5, wildtype × heterozygous matings produce normal Mendelian ratios (1:1 wt:het), whereas pups genotyped following weaning are approximately 63% wildtype and 37% heterozygous, further demonstrating mortality in *Gabra2*–1 prior to weaning, but after PND 5. We genotyped the offspring found dead from heterozygous × heterozygous mating pairs and found that when corrected for predicted offspring ratio (2:1 het:homo), heterozygous and homozygous offspring have similar likelihood of mortality, but homozygous mice appear to die earlier in development (mode: het = PND 20, homo = PND 17; Fig. 7b). A representative survival plot from a heterozygous × homozygous mating, again reflects the peak of mortality around PND 20 (Fig. 7c). We also compared the percentage of litters that had one or more dead pups, and the percentage of pups weaned comparing all mating schemes (shown on the *x*-axis; Fig. 7d, e). Dead pups were rarely found in wildtype × wildtype mating pairs and approximately 98% of wildtype pups born survive through weaning. In contrast, *Gabra2*–1 mating schemes had at least 69% of litters with one or more pups found dead after PND 5, and only about 36% of pups born survive through weaning in *Gabra2*–1 homozygous mating pairs.

We evaluated failure to thrive by assessing size, weight, and gross physical development of *Gabra2*–1 pups, finding no difference from wildtype (Supplementary Fig. 8A, B, Supplementary Table 4). Extensive monitoring did reveal the occurrence of rare spontaneous seizures in *Gabra2*–1 offspring, with clear behavioral characteristics that we classified according to the Racine scale (Fig. 7f; Supplementary Movie 1). To confirm this

apparent predisposition to seizure activity, we also assessed the susceptibility of adult *Gabra2*–1 mice to kainate (20 mg kg$^{-1}$ i.p.) induced seizures (Fig. 7g). Using the Racine scale we found that *Gabra2*–1 mice consistently showed more severe seizures and increased mortality resulting from kainate injection. EEG recordings also revealed increased sensitivity to kainate, with all

*Gabra2*–1 mice developing status epilepticus (SE—less than a 2-min interval between events), while only half of wildtype mice developed SE. The latency to the development of SE was reduced in *Gabra2*–1 mice (Fig. 7h, i), while latency to the first seizure event did not differ (Fig. 7h, j). *Gabra2*–1 mice also have seizure events of increased duration (Fig. 7h, k). Taken together,

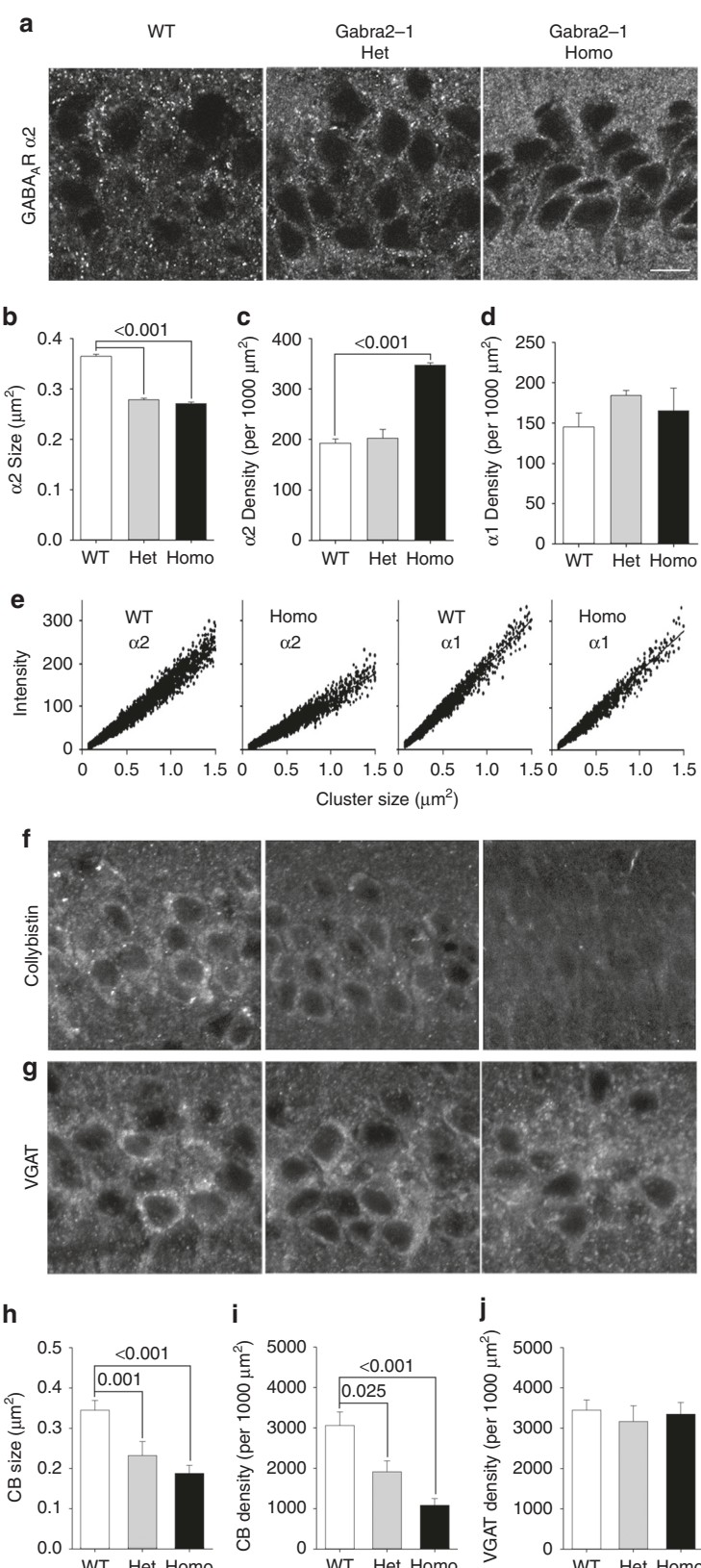

**Fig. 4** Expression of GABA$_A$R α1 and α2 subunits, and related inhibitory synaptic proteins in *Gabra2*–1 mice. **a** Representative images of GABA$_A$R α2-subunit staining in the CA1 region of the hippocampus. Quantification of the size: (**b** WT—0.365 ± 0.004; Het—0.279 ± 0.003; Homo—0.271 ± 0.003), and density (**c** WT—192.80 ± 7.851; Het—202.70 ± 17.490; Homo—347.20 ± 5.276) of α2 positive clusters in CA1. **d** Quantification of GABA$_A$R α1 subunit cluster density (WT—145.00 ± 17.50; Het—184.10 ± 6.31; Homo—165.00 ± 28.33) in CA1. **e** Cluster size/intensity correlation plots for α2 and α1 staining, showing the shift toward smaller, less intense clusters of α2 in *Gabra2*-1 homozygous mice. **f** Representative images of CB immunostaining in CA1. **g** Representative images of VGAT immunostaining in CA1. Quantification of the size (**h** WT—0.344 ± 0.024; Het—0.232 ± 0.035; Homo—0.188 ± 0.020) and density **i** WT—3056.06 ± 338.30; Het—1913.26 ± 275.02; Homo—1087.94 ± 165.43) of CB positive clusters in CA1. **j** Quantification of VGAT positive cluster density (WT—3447.82 ± 250.00; Het—3165.59 ± 390.82; Homo—3346.53 ± 289.30) in CA1. Scale bar = 10 μm, applies to all images. All plots shown and all values listed are mean ± standard error, *p* values from KS test (cluster size), or ANOVA (cluster density)

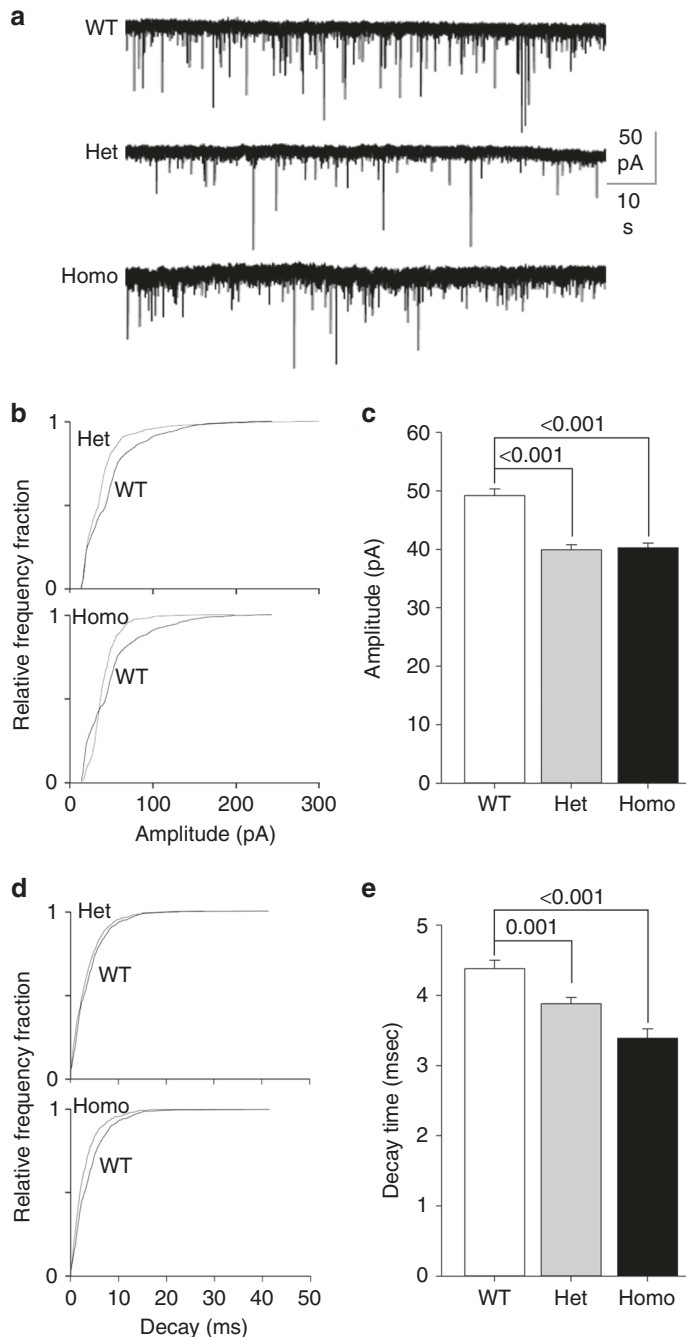

**Fig. 5** *Gabra2*–1 mice display alterations in hippocampal phasic current. **a** Representative recordings made from CA1 in hippocampal slices from p21 *Gabra2*–1 heterozygous and homozygous mice compared to wildtype littermate control mice. *Gabra2*-1 recordings displayed smaller sIPSC amplitudes and decreased decay times compared to wildtype littermate or age-matched controls as seen in the representative traces. Analysis of traces revealed a shift in the cumulative probability plots and bar graphs for sIPSC amplitude (**b**, **c** WT—49.2 ± 1.16; Het—39.9 ± 0.91; Homo—40.3 ± 0.74) and decay (**d**, **e**. WT—4.38 ± 0.12; Het—3.88 ± 0.09; Homo—3.39 ± 0.13). The frequency was comparable between the genotypes, for quantification see Supplementary Fig. 7. All plots shown and all values listed are mean ± standard error, *p* values from *t* test

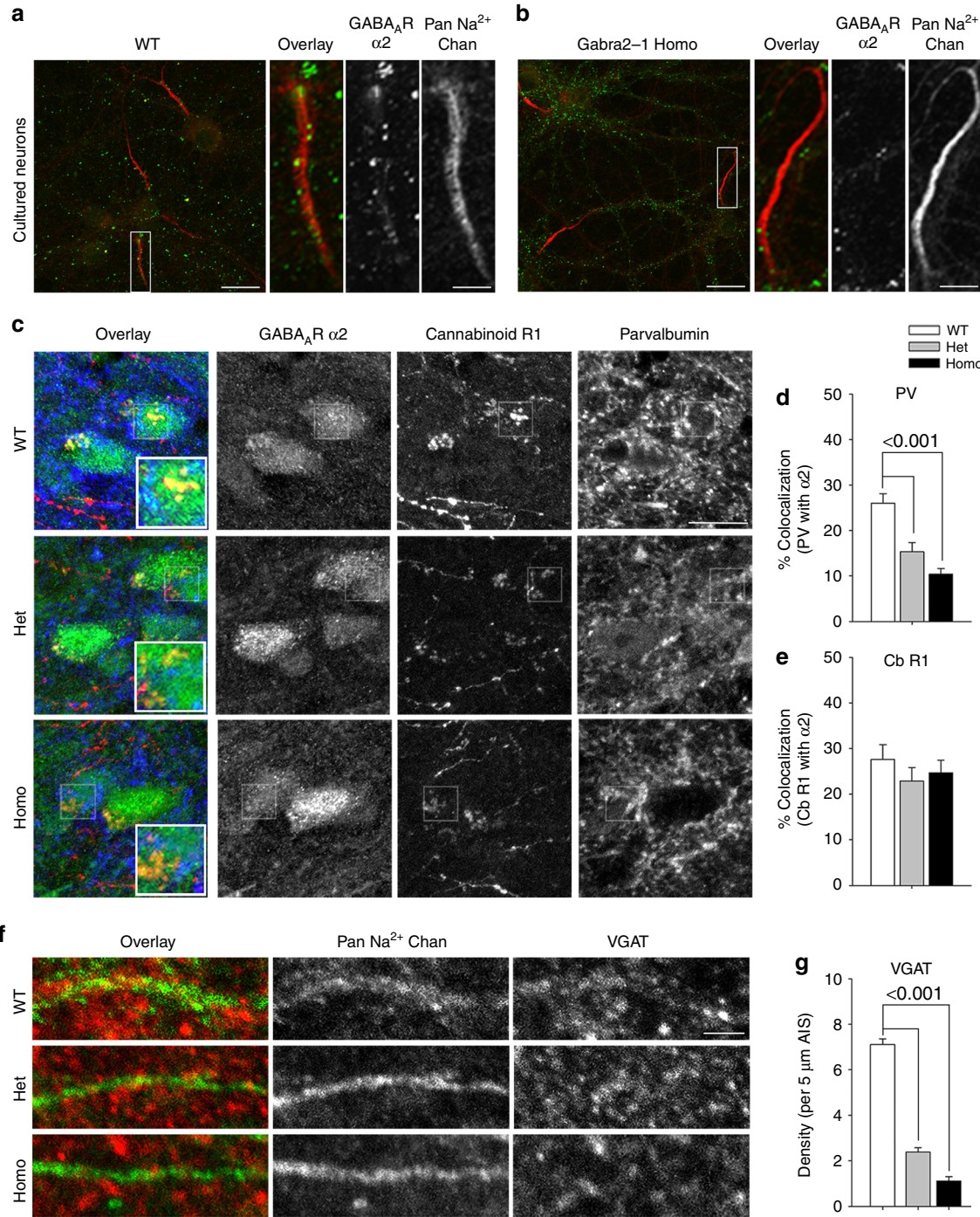

**Fig. 6** Examination of the subcellular localization of GABA$_A$R α2-subunit clusters in *Gabra2*-1 mice. Cultured cortical neurons from wildtype (**a**) and Gabra2-1 homozygous (**b**) pups stained with antibodies directed against GABA$_A$R α2 (green) and sodium channels (red), showing zooms of representative AIS segments. **c** Colocalization of GABA$_A$R α2 (green) with CB1R (red) and parvalbumin (blue) clusters in the CA1 of wildtype, heterozygous and homozygous Gabra2-1 mice. Quantification of the colocalization of GABA$_A$R α2 with parvalbumin (**d** WT—26.05 ± 2.12; Het—15.34 ± 2.06; Homo—10.41 ± 1.24) and CB1R (**e** WT—27.64 ± 3.30; Het—22.90 ± 2.99; Homo—24.73 ± 2.74). **f** Staining for VGAT (red) positive clusters to mark inhibitory presynaptic compartments, along with a Pan-sodium channel antibody (green) to label the axon initial segment. **g** Quantification of the number of VGAT positive clusters per 5 μm of AIS (WT—7.11 ± 0.26; Het—2.39 ± 0.19; Homo—1.11 ± 0.18). Scale bar = 10 μm (A low mag, C), scale bar = 5 μm (A zoom, F). All plots shown and all values listed are mean ± standard error, p values from ANOVA

we identified a susceptibility of *Gabra2*-1 mice to spontaneous and chemically induced seizures that can result in early mortality.

**Gabra2-1 anxiety and EEG corrected by α2-selective compound.** To explore the phenotype caused by *Gabra2*-1 mutation, we examined surviving adult *Gabra2*-1 mice using behavior tests

and EEG recordings. To rule out the possibility of gross behavioral impairment we conducted a modified SHIRPA screen[32] (Supplementary Table 4; Supplementary Fig. 8B). *Gabra2*-1 mice were shown to have normal physical health, weight, body condition, as well as sensory and motor function (Supplementary Table 4; Supplementary Fig. 8). Assessment of the startle response

to evaluate hyperekplexia, which is linked to glycine receptor function[33] and human mutations in *ARHGEF9*[24], revealed no significant difference between *Gabra2*–1 and littermate controls (Supplementary Fig. 8E, F). We also evaluated the possibility of a deficit in prepulse inhibition due to reports of mislocalization of the GABA$_A$R-α2-subunit on the AIS in post mortem tissue from schizophrenic patients[34], but no difference was detected between *Gabra2*–1 mice and littermate controls (Supplementary Fig. 8G, H). Next, we assessed anxiety behavior, due to the role of the GABA$_A$R-α2-subunit in mediating the anxiolytic effects of benzodiazepines[35]. Using both the light–dark boxes and elevated plus

maze, we detected increased anxiety behavior in *Gabra2*–1 mice shown by increases in total time spent in the dark chamber, and percent time spent in the closed arms respectively (Fig. 8a, b). Due to the low numbers of viable homozygotes we examined the responsiveness of *Gabra2*–1 heterozygotes to the nonselective benzodiazepine, diazepam (2 mg kg$^{-1}$), and the α2/α3-selective GABA$_A$R positive modulator AZD7325 (3 mg kg$^{-1}$). As expected, wildtype mice showed a reduction in anxiety behavior following treatment with either diazepam or AZD7325 as evidenced by an increased percent time spent in the open arms (Fig. 8c), and increased open arm entries (Supplementary Fig. 9A). In contrast,

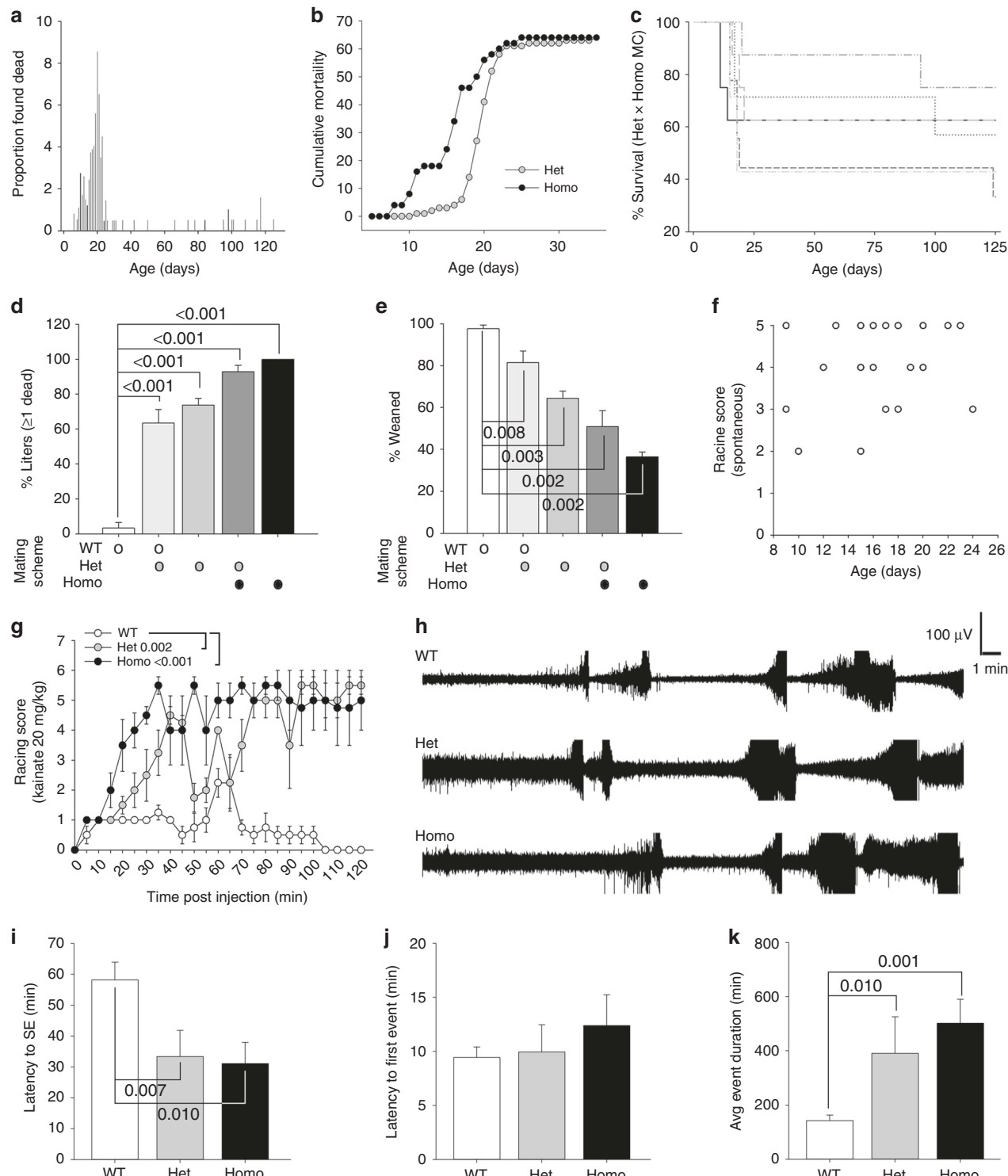

**Fig. 7** Early postnatal mortality and spontaneous seizures in *Gabra2*–1 heterozygous and homozygous mice. **a** Plot showing the proportion of mice found dead (out of the total found dead) across a 120 day lifespan. **b** Cumulative proportion of heterozygous and homozygous mice found dead in the early postnatal period, corrected for predicted Mendelian ratio of 2:1 het:homo. **c** Survival plot of litters from a typical heterozygous × homozygous mating pair. Quantification of the percentage of litters from different mating schemes that had one or more dead pups (**d** WT × WT—3.20 ± 3.20; WT × Het—69.05 ± 8.29; Het × Het 77.38 ± 4.49; Het × Homo 90.91 ± 9.09; Homo × Homo—100.00 ± 0.00) and the percentage of pups that were weaned (**e** WT × WT— 97.78 ± 2.22; WT × Het—60.63 ± 4.98; Het × Het 63.57 ± 4.42; Het × Homo 46.41 ± 14.13; Homo × Homo—36.50 ± 2.21) according to mating scheme. **f**. Quantification of spontaneous seizure severity according to the Racine scale in pups observed seizing. **g**. Quantification of kainate seizure severity according to the Racine scale in WT (LS Mean 0.74), heterozygous (LS Mean 3.40), and homozygous (LS Mean 4.19) littermates (SE of LS Mean = 0.401; Het vs Homo $p$ = 0.197). **h** Representative traces of EEG activity in wildtype, and *Gabra2-1* heterozygous and homozygous mice following kainate injection. Quantification of the latency to SE (**i** WT—58.16 ± 3.31; Het—33.39 ± 5.52; Homo—31.09 ± 4.43; Het vs. Homo $p$ = 0.732), latency to first seizure event (**j** WT—9.44 ± 0.973; Het—9.95 ± 2.523; Homo—12.39 ± 2.836), and the average event duration (**k** WT—143.08 ± 13.05; Het—390.71 ± 88.21; Homo— 501.92 ± 57.63; Het vs. Homo $p$ = 0.214) in wildtype, heterozygous, and homozygous *Gabra2-1* mice treated with kainate. All plots shown and all values listed are mean ± standard error, $p$ values from ANOVA, or repeated measures ANOVA (Kainate Racine over time)

*Gabra2*–1 mice did not show a reduction in anxiety behavior following diazepam treatment, but were responsive to treatment with AZD7325 (Fig. 8c, d; Supplementary Fig. 9A). Analysis of total distance traveled did not reveal any differences in exploration of the maze between genotypes, or treatment compared to vehicle control (Supplementary Fig. 9B). It is possible that *Gabra2*–1 mice require an increased dose of diazepam to exert anxiolytic effects; however, higher doses of diazepam cause sedation. As such, we also examined the responsiveness of *Gabra2*–1 mice to sedative doses of diazepam (10 mg kg$^{-1}$) to rule out any difference in drug metabolism. We found no difference in the sedative effects of diazepam comparing wildtype and *Gabra2*–1 mice (Supplementary Fig. 9C, D). These results suggest that *Gabra2*–1 mice display an anxiety phenotype possibly related to impaired clustering and function of α2-containing GABA$_A$Rs.

We next assessed adult *Gabra2*–1 mice for baseline abnormalities in EEG recordings. *Gabra2*–1 mice and controls were recorded during 1 h of wakefulness at least one week following implantation. Representative spectrograms and fast-Fourier analysis revealed increased power of specific frequency bands when *Gabra2*–1 mice and littermate controls were compared (Fig. 8e, f). To examine this more closely, we parsed the EEG into frequency bands as follows: δ (0.25-4 Hz), θ (4–12 Hz), α (12–18 Hz), β (18–30 Hz), and γ (>30 Hz), revealing an elevation particularly in δ and θ ranges in *Gabra2*–1 mice (Fig. 8g; Supplementary Fig. 10). To follow up on the findings with anxiety behavior, we investigated the beneficial effects of AZD7325 (3 mg kg$^{-1}$) in correcting the EEG abnormalities in *Gabra2*–1 (Fig. 8e, g). We found that AZD7325 was successful in reducing EEG δ and θ power in *Gabra2*–1 mice (Fig. 8e, g; Supplementary Fig. 10A–E). We also found that AZD7325 resulted in increases in the power of high frequency bands in both wildtype and *Gabra2*–1 heterozygotes (Supplementary Fig. 10F, G) similar to previous findings[36]. Diazepam is also known to suppress δ-power, so we conducted assessment of the response to diazepam following 1 h of baseline EEG, comparing wildtype to *Gabra2*–1. Diazepam suppressed δ in wildtype mice, but not in *Gabra2*–1 mice (Supplementary Fig. 10H), which is in keeping with the lack of behavioral response to diazepam (Fig. 8c, d). Collectively, the EEG analysis revealed abnormally elevated δ power in *Gabra2*–1 mice compared to wildtype controls, which could be alleviated by treatment with AZD7325 but not diazepam, mirroring the behavioral results.

## Discussion
We show that the CB-SH3 domain specifically binds the GABA$_A$R-α2-subunit loop with low micromolar affinity, which is the strongest interaction identified for both proteins to date.

Despite its high affinity, the interacting α2-subunit fragment is devoid of canonical PxxP motifs, indicating an atypical SH3 interaction. Earlier Y2H studies demonstrated that full-length CB and gephyrin together exhibit a potentiated GABA$_A$R-α2 affinity[19]. Our results exclude a possible gephyrin-SH3 domain interaction, as well as synergistic binding to the GABA$_A$R-α2-subunit in the presence of gephyrin. Instead we propose that Y2H-demonstration of gephyrin's ability to amplify CB's binding uncovered gephyrin-mediated un-masking of CB-SH3[37]. Isolated CB-SH3 therefore might exhibit the maximal affinity displayed by CB after full activation by gephyrin[37]. Comparison with gephyrin reveals a striking reciprocal relationship in CB-affinities with different α-subunits. A possible gephyrin–CB complex would, therefore, have the potential to interact with all three synaptic α-subunits, thus explaining reports of both CB and gephyrin colocalizing with GABA$_A$Rs containing α1–3-subunits. Therefore, our results indicate that the molecular basis of GABA$_A$R and, possibly, also GlyR accumulation at synaptic sites lies in the so far unknown architecture and modulation of the CB–gephyrin complex, as well as in its exact interplay with other binding partners (PI3P, neuroligin-2, and CDC 42).

Both GABA$_A$R-subunits[38–40] and CB[24–27] have been implicated in human disease. We have previously shown that GABA$_A$R signaling can be modulated by interfering with the underlying protein–protein interactions[41,42]. Here, we used a novel mouse expressing a chimeric α2–α1 loop to investigate the disease relevance of the CB-GABA$_A$R-α2 interaction. The *Gabra2*–1 mutation reduces GABA$_A$R-α2 interaction with CB in vitro, and results in an increase in α2, but a decrease in CB expression in vivo. The increase in α2 density may reflect an increased stability due to interaction with gephyrin or other partners in place of CB, or may reflect poor clustering of α2-containing receptors, which is suggested by decreased cluster size and reduced mIPSC amplitude. Although the *Gabra2*–1 mutation appears to reduce clustering of α2-containing receptors, we do not see a detectable reduction in the size or density of VGAT (or GAD-65) positive clusters, or when measured specifically on the soma, or in the proportion of VGAT clusters that have α2, suggesting no widespread loss of inhibitory synaptic contacts. These results suggest that the *Gabra2*–1 mutation is impairing the interaction of α2 with CB, leading to reduced clustering of α2-containing receptors and increased turnover of CB. CB-KO mice show a loss of α2-containing GABA$_A$Rs in CA1 pyramidal cells, accompanied by a loss of gephyrin and α1-containing GABA$_A$Rs[23]. In contrast, PV interneurons retain gephyrin and α1 expression upon CB KO[23]. This may suggest that CA1 pyramidal cells express heteromeric α1–α2 receptors, which rely upon CB–α2 for expression and retention at synapses, while other cells expressing GABA$_A$Rs with two α1-subunits are not impacted by the loss of CB–α2 interaction[43]. We also found that the *Gabra2*–1 mutation has a

significant effect on inhibitory synaptic currents recorded from CA1, reducing their amplitude and accelerating decay time. A decrease in decay time in *Gabra2*–1 slices likely reflects the loss of effective activation of α2-containing GABA$_A$Rs, which have been shown to display a 2–6 times slower deactivation[8,31].

As we could not detect a reduction in total GABAergic synapses (VGAT, GAD-65 clusters; colocalization of VGAT-α2) we next evaluated the effect of the *Gabra2*–1 mutation on specific subsets of synapses. We observed fewer α2 positive clusters along the AIS in *Gabra2*–1 cells compared to wildtype cells. We compared the colocalization of α2 with CB1R and PV in tissue sections, to differentiate between axo-somatic contacts and AIS contacts from chandelier cells. We found that the *Gabra2*–1 mutation caused a substantial loss of α2 at PV + sites, and no significant difference in α2 at CB1R + sites, suggesting an impact on AIS synapses. *Gabra2*–1 mice indeed displayed a significant

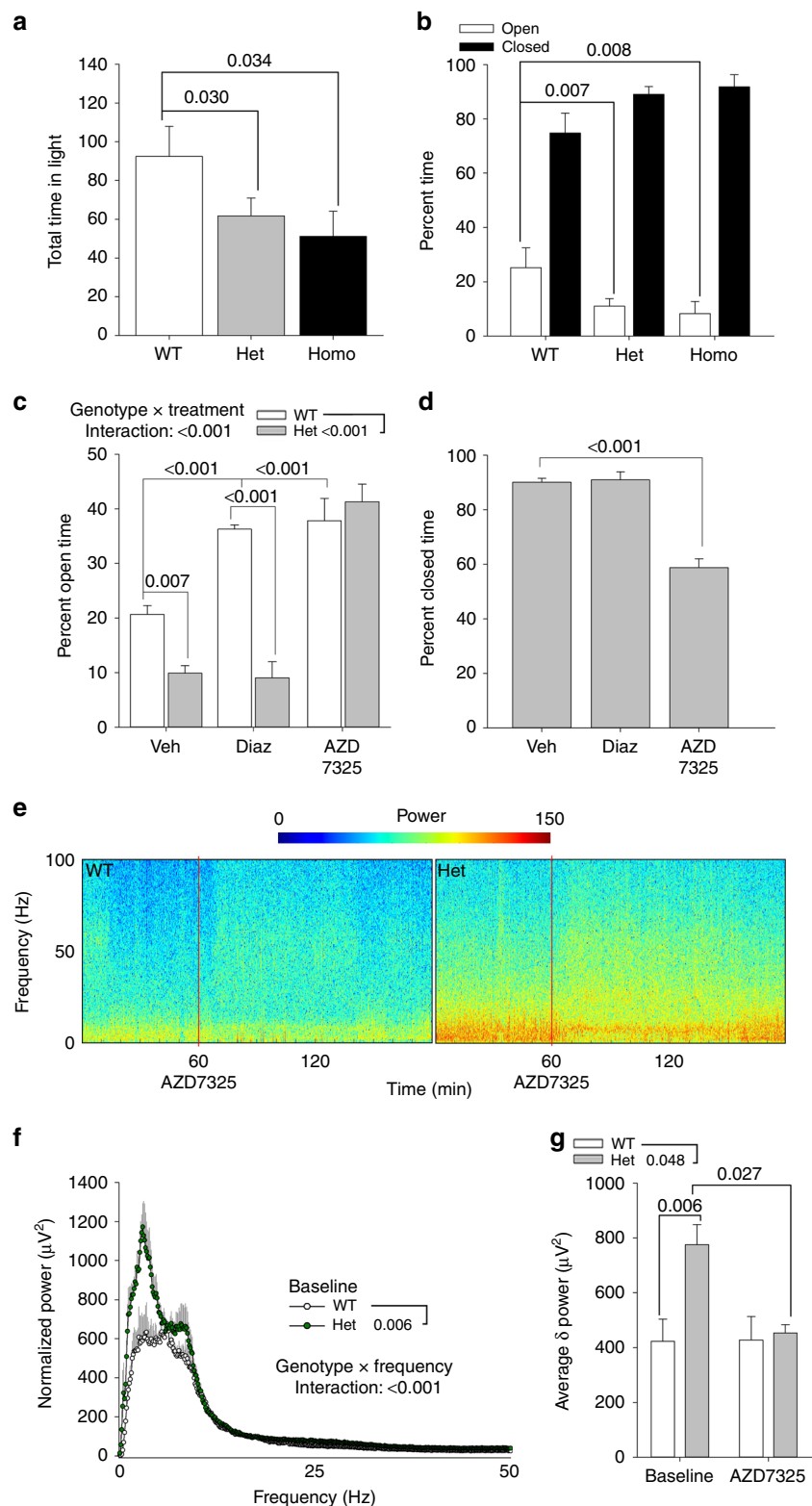

**Fig. 8** Anxiety and EEG in *Gabra2*–1 mice are corrected by treatment with an α2-selective compound. **a** Quantification of the percent time spent in the light chamber of the light-dark box, comparing wildtype (104.52 ± 11.11), heterozygous (71.77 ± 6.89) and homozygous (73.50 ± 10.34) *Gabra2-1* mice. **b** Time spent in the arms expressed as percent of total in the elevated plus maze comparing wildtype (open—25.22 ± 4.87; closed—74.78 ± 4.87), heterozygous (open—10.95 ± 1.90; closed—89.05 ± 1.89) and homozygous (open—8.23 ± 2.95; closed—91.78 ± 2.95) *Gabra2-1* mice. **c** Assessment of the percent time spent in the open arms of the elevated plus maze comparing wildtype (LS Mean 31.58) and Gabra2-1 heterozygous (LS mean 20.10; SE of LS mean for genotype—1.51) mice treated with vehicle (beta cyclodextran; LS mean 15.29), 2 mg kg$^{-1}$ diazepam (LS mean 22.67), or 3 mg kg$^{-1}$ AZD7325 (LS mean 39.55; SE of LS Mean for treatment—1.85). **d** Analysis of percent closed arm time in *Gabra2-1* heterozygous mice. **e** Spectrograms of representative EEG recordings from *Gabra2-1* heterozygotes and wildtype controls during baseline and after treatment with AZD7325. **f** FFT of EEG recordings comparing wildtype (LS mean 82.02 ± 8.23) and *Gabra2-1* heterozygous (112.60 ± 7.13) mice. **g** Comparison of average δ power between wildtype and *Gabra2-1* mice, and after treatment with AZD7325. All plots shown and all values listed are mean ± standard error, p values from ANOVA, or repeated measures ANOVA (EEG FFT)

reduction in the number of VGAT clusters opposed to the AIS, suggesting that other subunits do not compensate for the loss of α2. A recent study has questioned the selective segregation of α2, suggesting that α1-subunits are also enriched on the AIS[44]. The approach relied upon different polyclonal antibodies that have distinct affinities, hence an absolute quantification of subunit levels is not possible. Moreover, data interpretation is complicated by estimates that in excess of 20% of GABA$_A$Rs containing α2-subunits also contain α1[45,46]. Another recent study identified some region-specific expression of α1 and α3 on the AIS, by comparing AIS expression to non-AIS expression to overcome antibody affinity differences[47]. While α1 and α3 are detected on the AIS, α2 shows the greatest enrichment, and further gephyrin is not preferentially enriched and may not contribute to AIS recruitment of GABA$_A$R-α-subunits[47]. Our new findings suggest that α2 interaction with CB is the primary mediator of localization of GABA$_A$Rs to AIS synapses, and that other subunits and gephyrin may be localized there in small amounts when complexed with α2 and CB. The specific impact of *Gabra2-1* mutation on AIS inhibitory synapses may be related to interaction of the α2–CB complex with other proteins that have been implicated in the stabilization of AIS functional components such as ankyrin$_G$[48], neurofascin[49], NrCAM[50], and βIV-spectrin[51].

Stemming from these alterations to GABAergic synapses and inhibitory synaptic function, *Gabra2-1* mice display a remarkable phenotype characterized by spontaneous seizures, resulting in mortality in a subset of both heterozygous and homozygous mice. This is particularly striking since a relatively small number of GABA$_A$R-subunit mutations result in spontaneous seizures[38–40,52–54]. A role for AIS abnormalities in epilepsy has been suggested in other studies[55–57], implicating trafficking and localization of voltage-gated channels to the AIS. Other analyses have found alterations to AIS inhibitory synapses resulting from or contributing to epilepsy; however, both decreases and increases in AIS synapse number have been found, leaving the contribution of this synaptic subtype unclear[58–60]. Our results directly link a mutation in a specific GABA$_A$R-subunit, accompanied by an alteration in AIS GABAergic synapses, to an epileptic phenotype. An additional feature of the *Gabra2-1* model is that it reduces the possibility of gene compensation relative to a subunit knock-out. This is supported by the fact that spontaneous seizures and early mortality are not reported in the α2[61] or CB[21] knock-out models where a compensatory feedback enhancing the expression of other proteins may occur.

Surviving *Gabra2-1* mice show anxiety-like behavior and alterations in EEG, including an enhancement of δ-power. *Gabra2-1* anxiety may be related to prior seizures, as it is well documented that anxiety is a common comorbidity in human epilepsy and in animal models[62]. The alterations in δ-power also corroborate the anxiety findings. δ-frequency oscillations have been most closely tied to salience detection[63,64], and anxiety as a heightened salience state is correspondingly linked to elevations in the δ-range[65]. Further, nonsedating doses of diazepam and

other benzodiazepines are known to decrease δ-power[66,67]. While AZD7325 reduced anxiety-like behavior in both *Gabra2-1* mice and wildtype controls, we found that the classical benzodiazepine diazepam was not effective at reducing anxiety-like behavior in *Gabra2-1* mice. Further, when we analyzed the impact of diazepam on the EEG, we also found that *Gabra2-1* mice do not show the typical suppression of δ. Previous studies have suggested that binding of benzodiazepines to α2-containing receptors is required for the suppression of EEG δ-power[68], while α1 and α3 binding has a limited role in δ-suppression[69,70]. Although it is not immediately apparent why diazepam is ineffective in *Gabra2-1* mice, this difference may be related to the predominance of α1-/α3-mediated effects of diazepam when α2-containing receptors are compromised. In contrast, since AZD7325 acts preferentially on α2-containing receptors, it remains effective at suppressing δ and reducing anxiety-like behavior.

The *Gabra2-1* mutation demonstrates a striking parallel with the effects of *ARHGEF9* mutation in human syndromes, whose symptoms include epilepsy, anxiety, mental retardation, aggressive behavior, sleep–wake cycle disruptions, early mortality, and hyperekplexia[24–27]. It is likely that the hyperekplexia is linked to CB interaction with glycine receptors[15], however, the remaining symptoms may be linked to dysfunctions at the AIS due to mislocalization of α2-containing GABA$_A$Rs. *Gabra2-1* mice do not display hyperekplexia, but have increased seizure susceptibility, early mortality, and anxiety, drawing several parallels. The responsiveness of the anxiety and EEG phenotype to the α2/α3-selective GABA$_A$ positive modulator AZD7325 suggests a possible novel therapeutic strategy to relieve symptoms in this set of human developmental syndromes linked to CB dysfunction.

Taken together, our results reveal a clear and selective interaction between the α2-subunit of GABA$_A$Rs and CB, an interaction that has been long speculated about but has not previously been characterized. Furthermore, using the novel *Gabra2-1* mouse model, we demonstrate that this interaction is critical for the targeting of α2-containing GABA$_A$Rs, specifically to AIS sites needed for the stabilization of VGAT positive contacts onto the AIS. *Gabra2-1* mice display spontaneous seizures during development, accompanied by early mortality in a subset of pups. Adult *Gabra2-1* mice that survive have increased anxiety-like behavior and EEG abnormalities that are responsive to α2-selective benzodiazepines, revealing a novel therapeutic potential for these compounds.

## Methods

**Protein expression and purification**. GephE (AA 318–736) as well as the large cytoplasmic loop of the GlyR β (AA 378–425), GABA$_A$R α1 (AA334–420), α2 (AA 335–419) and α3 (AA 392–406) subunits and the corresponding truncation variants GABA$_A$R α2 (AA 307–334), and (AA 307–350) were expressed and purified. The pGEX Vector SH3 construct provided by T. Soykan from the Brose Lab corresponds to residues 4–72 of murine CB1 and encompasses the complete SH3 domain (residues 8–67) of this protein which is identical to the SH3 domain of the human CB isoform, hPEM-2. The protein was expressed in *Escherichia coli* and purified via its N terminal GST-tag. After in-situ thrombin cleavage the GST–SH3

mixture was applied to a 26/60 Superdex 200 size exclusion column (Amersham Biosciences) equilibrated with 10 mM Tris-HCl pH 8.0, 250 mM NaCl, 1 mM β-mercaptoethanol. Two separate peaks could be resolved, one containing both the cleaved and uncleaved GST and GST–SH3, respectively, and the other containing pure SH3 domain.

**Peptide synthesis.** Peptides were purchased as lyophilized powder from Genscript (USA) at a minimal purity of 90%.

**GFP bead trap and western blotting.** HEK293T cells were transfected by electroporation with 2 μg DNA of each construct, and used for experiments 48 h later. HEK cells were transfected with full-length CB and either phluorin-tagged full-length α2 or phluorin-tagged full-length α2–1 constructs. Cultures were chilled on ice and lysed in 20 mM Tris-HCl (pH 8), 150 mM NaCl, 1% triton, 5 mM EDTA, 10 mM NaF, 2 mM Na$_3$VO$_4$, 10 mM Na$_4$P$_2$O$_7$ with protease inhibitors. Lysates were cleared by centrifugation and incubated with GFP-trap overnight at 4 °C. After four washes in lysis buffer, bound proteins were detected by western blotting.

A table of antibodies, their sources and concentrations used can be found in Supplementary Table 3. For western blotting, HEK cells or brain tissue samples ($n$ = 6 mice per genotype in >2 independent experiments) were rapidly harvested and homogenized in TEEN buffer (50 mM Tris-HCL, 1 mM EGTA, 150 mM NaCl) supplemented with a protease inhibitor mixture (Roche Applied Science). Following homogenization the protein concentration was assessed using a BCA assay (Pierce) and heated to 65 °C in SDS-PAGE sample buffer with 10% β-mercaptoethanol for 15 min. After SDS-PAGE, western blot signals were detected using ECL. For quantification, specific protein levels were determined by densitometry and were normalized to actin loading densitometry results. Results were quantified using Image J and were analyzed using $t$ test (bead trap) or analysis of variance (ANOVA) (western blotting) to compare mean values, graph plotted as mean ± standard error.

**Mouse generation and maintenance.** Animals were cared for according to the NIH Guide for the Care and Use of Laboratory Animals[71] and protocols were approved by the Institutional Animal Care and Use Committee (IACUC) of Tufts University School of Medicine or the IACUC of the University of Nevada Las Vegas. Germ line transmission of the transgenes was detected using PCR with primers spanning the intronic region which contained the remaining loxP site (Supplementary Fig. 1). Nontransgenic littermates were used as controls in all experiments (wildtype). Following genotyping, animals of each genotype were randomly assigned to treatment and/or assessment groups. Animals were generated by Genoway and maintained at the vivarium for Tufts University's Boston campus, as well as at the University of Nevada Las Vegas vivarium. Mice were group housed with a 12 h light–dark cycle with constant temperature. After founders were received, mice were back-crossed onto C57 Bl6J for more than ten generations (congenic) prior to behavioral and EEG assessments. For tissue IHC, behavior, pharmacology, and EEG experiments, only male mice, aged 6–10 weeks, were used. All other experiments included both males and females (western blotting, culture immuno (cultures made a P0), electrophysiology (slices cut at P18-22), and assessments of mortality).

Genotyping Primers:
Gabra2-1F: AAGGGGAATTGTTTGGAGCAAGTGAC
Gabra2-1R: TCCACTCGCCTTATGGTGAACTTATAAGG
Sequencing Primers:
α2se1: GCGCCGGTATTGTTCTCTG
α2se2: TGACCCCTAATACAGGCTCCC
α2se3: ATGCAGCCTAAACCCACATTG
α2se4: GCTTCTTGTTCGGTTCTGGC

**Immunocytochemistry and IHC.** A table of antibodies, their sources and concentrations used can be found in supplementary Table 3. For immunocytochemistry (ICC) and IHC, four preservation protocols were used ($n$ = 4 coverslips per genotype ICC; $n$ = 6 mice per tissue preparation method IHC). (1) For ICC, at day in vitro 14 (DIV 14) coverslips of cultured cortical cells were fixed in −20 °C methanol for 10 min. (2) For nonfluorescent IHC of GABA$_A$ receptor α2-subunits, animals were transcardially perfused with a periodate lysine paraformaldehyde fixative solution. (3) For fluorescent IHC of GABA$_A$R-α1- and -2-subunits, animals were transcardially perfused with oxygenated artificial cerebrospinal fluid followed by post fixation in 4% paraformaldehyde in PB. (4) For fluorescent IHC of CB, VGAT, colocalization of α2 subunit with parvalbumin and cannabinoid receptor 1, and VGAT/sodium channel experiments, brains were harvested freshly and flash frozen in optimal cutting temperature embedding medium in liquid nitrogen cooled isopentane. Following one of the above preparation methods, sections were cut on a cryostat at thicknesses of 10 μm (fresh frozen—mounted directly on slides, fixed in −20 °C methanol) or 30 μm (paraformaldehyde fixed—free floating) and were incubated in blocking solution (2.5% Bovine Serum Albumin, 5% Normal Goat Serum, 0.1% triton-x, and 0.02% sodium azide in PBS) for 45 min, followed by primary antibody incubations diluted in modified blocking solution (2% Normal Goat Serum) overnight at 4 °C, and biotinylated (Vector) or Alexa conjugated (ThermoFisher) secondary antibody

incubations diluted (1:2000) in modified blocking solution for 1 h at room temperature. Immunostaining intensity, density and colocalization was quantified using ImageJ, and results were analyzed using ANOVA to compare mean values, graphs were plotted as mean ± standard error.

**Electrophysiology.** Brain slices were prepared from 3-week-old male Gabra2–1 mice ($n$ = 6–13 cells/genotype, slices prepared from 5–6 mice and assessed independently, one mouse sliced per day) in ice-cold cutting solution (mM: 126 NaCl, 2.5 KCl, 0.5 CaCl$_2$, 2 MgCl$_2$, 26 NaHCO$_3$, 1.25 NaH$_2$PO$_4$, 10 glucose, 1.5 sodium pyruvate, and 3 kynurenic acid), and measurements of sIPSCs in CA1 principal neurons were performed. Slices were then transferred into incubation chamber filled with prewarmed (31–32 °C) oxygenated artificial cerebrospinal fluid of the following composition (mM: 126 NaCl, 2.5 KCl, 2 CaCl$_2$, 2 MgCl$_2$, 26 NaHCO$_3$, 1.25 NaH$_2$PO$_4$, 10 glucose, 1.5 sodium pyruvate, 1 glutamine, 3 kynurenic acid, and 0.005 GABA bubbled with 95% O$_2$–5% CO$_2$).

Whole-cell currents were recorded from the hippocampal CA1 region using patch pipettes (5–7 MΩ) pulled from borosilicate glass (World Precision Instruments) and filled with intracellular solution (mM: 140 CsCl, 1 MgCl$_2$, 0.1 EGTA, 10 HEPES, 2 Mg-ATP, 4 NaCl and 0.3 Na-GTP; pH = 7.2 with CsOH). A 5 min period for stabilization after obtaining the whole-cell recording conformation (holding potential of −60 mV) was allowed before currents were recorded using an Axopatch 200B amplifier (Molecular Devices), low-pass filtered at 2 kHz, digitized at 20 kHz (Digidata 1440A; Molecular Devices), and stored for off-line analysis. Spontaneous inhibitory postsynaptic currents (sIPSCs) were analyzed using Synaptosoft minianalysis software, and were analyzed using $t$ test to compare mean values, graphs were plotted as mean ± standard error.

**Electroencephalography.** EEG and electromyography electrodes were implanted under ketamine and xylazine anesthesia ($n$ = 6 mice per genotype per experimental agent—kainate and AZD7325). After a minimum of 5 d of postoperative recovery, EEG activity was measured using the Pinnacle system for mouse, during the dark phase of the cycle. For kainate experiments, mice were acclimatized to the recording chamber and preamplifier for one hour, then recorded for one hour of baseline before receiving 20 mg kg$^{-1}$ kainate i.p. The animals were then recorded for an additional 2 h postinjection. For baseline and AZD7325 experiments, mice were acclimatized to the recording chamber and preamplifier for 1 h, then recorded for 1 h of baseline before receiving 3 mg kg$^{-1}$ AZD7325 (or 40% (2-hydroxypropyl)-β-cyclodextrin vehicle control) i.p. For these experiments, vigilance was further promoted by providing animals with a fresh nestlet just prior to initiation of recordings. Thus, during the recording period mice were engaged with nest building, grooming, eating and drinking behaviors, and if mice became inactive a series of random tones was played. Results were quantified using SleepSign, and MatLab, and were analyzed using ANOVA (two way—Kainate and AZD7325 average experiments; repeated measures—FFT spectral analysis) to compare mean values, graphs were plotted as mean ± standard error.

**Behavioral assessments.** Genotype-blinded behavioral assessments were conducted on Gabra2–1 and wildtype mice during the light phase. For behavioral scoring of seizures, mice were assessed based on a Racine scale modified for mice, with 0 = normal behavior/no change; 1 = behavioral arrest, shivering; 2 = head nodding, Straub tail; 3 = forelimb clonus, champing; 4 = forelimb clonus with rearing; 5 = generalized tonic clonic, wild jumping; animals were assigned a 6 upon death. The preliminary screen was based on the modified SHIRPA protocol used by European Mouse Phenotyping Resource of Standardized Screens (EMPReSS) designed to evaluate the basic phenotype of transgenic mouse strains ($n$ = 8 mice per genotype run in >2 independent cohorts). The open field apparatus was based on that used in the EMPReSS resource. Open field behavior of mice was assessed using the Noldus Ethovision Tracking system from video recordings taken from above. To assess the sedative effects of diazepam, naïve mice were first habituated to the open field for 30 min, and then removed and dosed with 10 mg kg$^{-1}$ diazepam i.p. before being returned to the open field for an additional 60 min.

Protocols for both the light/dark exploration test and the elevated plus maze ($n$ = 8 mice per genotype per treatment run in >2 independent cohorts) were based on standard protocols. The light/dark arena is composed of a larger chamber that is transparent and brightly illuminated from above, and a smaller chamber that is black walled and dark. The two chambers are separated by a partition with a small doorway to allow the animal to freely pass from chamber to chamber while exploring for 10 min. The elevated plus maze was composed of two open and two closed arms extending from a central platform, elevated to a height of 40 cm above the floor. Test sessions were initiated by placing individual animals in the center square facing an open arm, and were terminated after 5 min of free exploration. For light-dark and elevated plus, video recordings (30 fps) of test sessions were digitized and both manually assessed by a genotype-blinded observer and analyzed using Noldus Ethovision, to quantify number of transitions between the light and dark compartments, and the total time spent in the light and dark compartments (light–dark), or the number of entries (all four paws into an arm) and time spent in open and closed arms (elevated plus). To test the anxiolytic effects of diazepam (2 mg kg$^{-1}$, 15 min prior to testing) or AZD7325 (3 mg kg$^{-1}$ in 40% (2-hydroxypropyl)-β-cyclodextrin, 45 min prior to testing) animals were given an ip

injection prior to initiation of the elevated plus maze, and compared to 40% (2-hydroxypropyl)-β-cyclodextrin vehicle controls. Results from all behavioral assessments ($n = 8$ mice per genotype, per treatment, run in 2–3 independent cohorts) were analyzed using ANOVA to compare mean values.

**Data collection and analyses.** Power analyses run on pools of data from independent drug treatment or genetic modification experiments reveal a range of sample sizes between 5 and 12 animals (with $n = 5$ animals per group yielding a power of 80% on simple tests, and $n = 10$ yielding a power close to 100%). For the purposes of the behavioral assessments within this set of studies, we estimated that eight animals would be required per genetic manipulation, and per drug group, per behavioral experiment.

Biochemistry, EEG and confocal imaging experiments are subject to somewhat less variability, and thus we estimated that fewer animals ($n = 6$) would be needed to obtain statistical power for these experiments. For electrophysiology, power analysis of prior data sets provided an estimate of five to six animals needed per group to achieve a power close to 100%. All findings were reproduced in >2 independent experiments (biochemistry, confocal microscopy, electrophysiology) or matched cohorts (EEG and behavior). Data from independent experiments or cohorts were compared to ensure that findings were reproduced, and then the data was combined for statistical analysis. Data were normally distributed, and data being compared had similar variance. No data were excluded from this manuscript. Investigators were blinded to genotype and/or treatment group for all confocal, electrophysiology, behavioral and EEG experiments, and subsequent analyses.

**Data availability.** The data that support the findings of this study have been deposited in an institutional repository, and are available from the primary author upon reasonable request.

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

## Acknowledgments

This work was supported by NIH-NINDS Grants NS101888, NS081735, and NS087662; NIMH Grants MH097446 and MH106954 (S.J.M.) and NIH-DA Grant DA037170 to S.J. M. S.J.M. serves as a consultant for SAGE Therapeutics and AstraZeneca, relationships that are regulated by Tufts University. This work was also supported by the Deutsche Forschungsgemeinschaft grants FZ 82, SFB 487 C7, SCHI 425/8-1 awarded to H.S. R.M. H. was supported by a CIHR postdoctoral fellowship. H.M.M. was supported by a grant of the German Excellence Initiative to the Graduate School of Life Sciences, University of Würzburg.

## Author contributions

R.M.H. generated and maintained the mice, performed genotyping, western blotting, perfusions, immunohistochemistry (cultured cells and fresh frozen tissue), confocal microscopy, assessments of mortality, seizure susceptibility, behavioral assessments, and performed data analysis on the aforementioned experiments, as well as assisted with EEG implants, recordings, and data analysis. H.M.M. expressed and purified recombinant proteins and carried out the native PAGE, analytical size exclusion experiments and isothermal titration calorimetry experiments. D.J.H. assisted with seizure susceptibility, and behavioral assessments, and conducted EEG implantation, recording, and analysis. P. P. and J.M.F. carried out the immunohistochemistry (perfusion fixed tissue) and analysis. A.M., T.D., and P.D. performed electrophysiological assessments. Y.N. conducted the GST-pull down experiment. A.J.N. assisted with genotyping and amplification of GABRA2 for sequencing. A.C. and N.B. contributed to the experiments examining diazepam and AZD7325. Figures for the manuscript were made by R.M.H. Experiments were designed by R.M.H., H.M.M., H.S., and S.J.M. The manuscript was written by R.M. H. with input from H.M.M., H.S., and S.J.M. S.J.M. and R.M.H. conceived the project.

## Additional information

**Competing interests:** S.J.M. serves as a consultant for SAGE Therapeutics and AstraZeneca, relationships that are regulated by Tufts University. The remaining authors declare no competing interests.

