## [Peer Review File · Nature Communications]

Reviewers' comments:

Reviewer #1 (Remarks to the Author):

This paper describes a novel and unique high affinity interaction between the GTP exchange factor collybistin and a cytoplasmic loop peptide of the $\alpha 2$ subunit of GABAARs, along with a putative role of this interaction in the synaptic accumulation of these receptors in the postsynaptic membrane. The authors had previously described this same $\alpha 2$ sequence as the site of interaction with the subsynaptic scaffolding protein, gephyrin. Others had shown evidence for a trimeric complex between the $\alpha 2$ subunit, gephyrin and collybistin in yeast-two-hybrid assays. In this paper here, the authors used detailed analyses of gephyrin- $\alpha 2$ and collybistin- $\alpha 2$ interactions with purified proteins in vitro to show that gephyrin and collybistin compete for interaction with the $\alpha 2$ subunit, rather than forming a trimeric complex. Mutation of the gephyrin/collybistin binding site of the $\alpha 2$ subunit by gene targeting in mice provides evidence that this interaction is critically important for accumulation of $\alpha 2$ receptors at synapses and also for stability and synaptic localization of collybistin. Behaviorally the mutant mice show increased mortality and spontaneous behavioral and EEG seizures, along with an anxiety phenotype that is reversible with the $\alpha 2$ specific benzodiazepine, AZD7325, but not with the broadly active benzodiazepine, diazepam. This manuscript for the first time provides a plausible mechanism for the differential synapse-specific accumulation of different subtypes of postsynaptic GABAARs that differ in their alpha subunit. The study is overall very well-conceived and comprehensive, and the experiments satisfy a very high standard. It should be of interest to the broad readership of Nature Communications. My concerns are relatively minor and they should be readily addressable.

Main concerns:

The observed loss of CB staining in Gabra2-1 neurons is surprising and unexplained. To differentiate between loss of CB accumulation due to loss of interaction with receptors and due to degradation of CB it would be important to test whether loss of CB is evident by western blot and whether this phenotype is replicated in $\alpha 2$ KO mice.

Fig 1 I,J shows a relatively modest loss of interaction between mutant Gabra2-1 cytoplasmic loop fragment and CB that is difficult to reconcile with virtually complete loss of mutant receptors from AISs in neurons. It would seem important to know whether loss of interaction between Gabra2-1 and CB is also evident and perhaps even more dramatic in the context of intact heteromeric ($\alpha 2\beta 2$) receptors.

As indicated by the authors in the Discussion, unaltered VGAT staining along pyramidal cell bodies of Gabra2-1 mice cannot be taken as evidence that somatic synapses are unaffected, and indeed this is one significant aspect of this manuscript that is unresolved. Given that loss of CB and $\alpha 2$ immunoreactivity is not limited to AIS synapses it seems likely that defects in (or altered) innervation are not limited to AIS synapses either. If VGAT+ synapses are unchanged and PV+ synapses are reduced (as evidenced at AISs), then one might predict a compensatory increase in somatic synapses from CCK basket cells, which the authors could quantitate by means of postsynaptic dystroglycan clusters.

Fig 7C, D. To ascertain the anxiety phenotype and anxiolytic-like effect of AZD7325, the authors should report % open arm entries and # of total arm entries or distance traveled. Please report all parameters for both genotypes.

The selective anxiolytic-like effect of AZD7325 vs diazepam of het mutants in the EPM is interesting but unexplained. Why would a reduction in AZD7325 targets result in increased behavioral response to AZD7325 but not diazepam – this should be discussed. Please add information on statistics including n, F and p values, and the type of tests used, to the text or figure legends for all experiments throughout the manuscript.

Other Details:

The text describing Fig 1C says that “a3L did not alter the elution profile of the SH3 domain”. To my eyes the elution profile of SHR in the presence of a3L looks the same as that in the presence of a1L. What am I missing? And how was the affinity of the a2L-SH3 complex estimated to be in the micro molar range, as part of Fig 1C?

In Suppl Figure 1B it would be helpful to add the Gabra1 sequence for comparison. Legend of Fig 1 reads “... complete saturation of GephE with the GABAAR α 2 subunit upon incubation with an 8-fold excess of the receptor loop.” Where does this 8-fold ratio come from?

What exactly does “expressed CB” refer to? Do you mean CB-containing crude extracts? The text on page 9 referring to Fig 2B-D and the corresponding Figure Legend need to be carefully revised with reference to the proper parts of this Figure

The text indicates that a2 expression in basal forebrain and hippocampus of Gabra2-1 mice is increased. Please include this data in Fig 2G. The text indicates that expression was increased in the molecular layer but this info is missing from Fig 2G. Presumably, the data labeled CA1 should be labeled pyramidal layer?

What were the genotypes of the mice quantified in Fig 6A-C? Can the data be reported separately for heterozygous and homozygous mutant offspring? Fig 6D, E are unclear also – it looks like the genotypes denoted along the X-axis refer to the parental mice but this is not stated in the Figure legend. Again, this data would be more informative if the data were disaggregated by genotype. Similarly, in Fig 6G it would be interesting to know whether heterozygotes were different from homozygotes.

Data in Fig 7 F are difficult to read. I suggest using a color code for genotype.

Reviewer #2 (Remarks to the Author):

This manuscript reports an interaction of the GEF collybistin (CB) with the intracellular loop of the GABAAR α 2 subunit and synaptic, electrophysiological and behavioural changes in line

with decreased GABAergic inhibition observed upon replacing the putative CB binding site in the mouse $\alpha 2$ gene by the respective GABAAR $\alpha 1$ sequence. As the mechanism of CB mediated gephyrin and GABAAR clustering is still enigmatic, this is basically an important result. However, after looking at the data provided I have considerable doubts about the authors' conclusions.

First, the binding data presented in Fig. 1 and Suppl. Table 2 are not in agreement with each other. The NAGE results showing comparable binding of CB's SH3 domain to both $\alpha 1L$ and $\alpha 2L$ are difficult to reconcile with a >500-fold difference in K_d determined by ITC. Also, the $\alpha 3L$ peptide appears not an adequate control, as it is considerably shorter and not as hydrophobic as the $\alpha 1L$ and in particular the $\alpha 2L$ sequences. Furthermore, all these quantitative interaction experiments are performed with the isolated SH3 domain and not recombinant CB. Second, no evidence for specificity of CB binding to the full-length $\alpha 2$ subunit is provided. Co-immunoprecipitation experiments as shown in Fig. 1I for $\alpha 2$ and $\alpha 2-1$ should also be performed for the full-length $\alpha 1$ and $\alpha 3$ subunits and ideally be complemented with immunoprecipitations from brain extracts in order to corroborate that CB binding is indeed $\alpha 2$ specific and occurring *in vivo*.

Another major concern relates to the central $\alpha 1$ sequence substituted into the *Gabra2-1* mice. In ITC, the corresponding $\alpha 2$ peptide ($\alpha 2$ min2) shows no significant binding to the SH3 domain, and the difference in CB intensity co-precipitated with the purified $\alpha 2$ and $\alpha 2-1$ subunits (Fig. 1I) is only small (about 30%; see Fig. 1J). Hence one wonders why an exchange of this motif was chosen for generating the *Gabra2-1* mice; I honestly have difficulties to understand the authors' strategy here. One argument they mention is conformation dependence of CB binding. However, I have not been able to detect evidence in support of this idea; rather, the fact that binding experiments can be performed with soluble tagged loop peptides seems contradictory.

The analysis of the *Gabra2-1* mice is overall straightforward and presents many results consistent with reduced synaptic inhibition. The immunocytochemistry presented in Fig. 3 and Suppl. Fig. 2, however, poses a number of questions. As noted by the authors themselves, in *Gabra2-1* sections $\alpha 2$ staining is much more diffuse than in WT specimens, suggesting (but not showing) that a major fraction of the $\alpha 2$ receptors is not synaptically localized. This should be evaluated by providing colocalization, and not just density, values for $\alpha 2$ and VGAT immunoreactivities, as done in Fig. 5. Ideally, colocalization values should also be determined for $\alpha 2$ and gephyrin both in sections and the AIS, since gephyrin constitutes a major binding partner of CB and has been shown to be required for $\alpha 2$ synaptic localization *in vivo*. Hence the loss of CB clusters could be a consequence of reduced gephyrin clustering at synaptic sites. The authors' proposal that CB might be eventually degraded in the *Gabra2-1* mice could be substantiated by immunoblots of WT and mutant brain extracts.

Another point that should be considered is the strong overexpression of the $\alpha 2-1$ subunit seen in at least some of the *Gabra2-1* mice (Fig. 2E, F). This could result in dominant-negative effects which might contribute to the electrophysiological and behavioural deficits described. For this reason, it also appears mandatory to analyze a sufficient number of

animals electrophysiologically. The mean values presented in Suppl. Fig. 3B are derived from only 2-3 animals. In my opinion, this is not sufficient for a reliable statistical analysis. Also, it appears desirable that precise animal n numbers are given for the quantitative results presented in several of the main figures.

In summary, although this study contains a plethora of significant data, I remain to be convinced that the statement made in the title of this manuscript is indeed correct. I therefore do not comment in detail on the other data presented on the survival, behaviour and pharmacological rescue of the Gabra2-1 mice, which are indicative of deficient GABAAR $\alpha 2$ neurotransmission.

Reviewer #3 (Remarks to the Author):

The manuscript "Developmental seizures and mortality result from reducing the affinity of the GABAA Receptor $\alpha 2$ subunit for collybistin" by Hines et al., investigates the molecular mechanisms of the GABAAR subtype localization in specific synapse subsets. In particular the Authors identify a high affinity binding between the scaffold protein collybistin (CB) and the alpha2 subunit of GABAA receptors that is responsible for the localization of alpha2-containing GABAA receptors in the axon initial segment, a sub-cellular compartment that is crucial for the control of neuronal excitability. In line with this, the Authors demonstrate that KI mice bearing a chimeric alpha2 subunit with disrupted CB binding show increased susceptibility to seizures, anxiety and early mortality.

The findings of the present work are very relevant. Indeed, this manuscript is shedding light on the mechanisms responsible for the clustering of GABAA receptor subtypes at specific synapses and/or sub-neuronal compartments, a subject that remains poorly understood. In addition the functional implications of the alpha2-CB interaction are shown at behavioral level thus linking malfunction of the synaptic function at the molecular level to the possible etiology of severe neurological disease.

This study is well conducted and employs a state-of-the-art multi-level approach that encompasses sophisticated biochemical techniques, electrophysiology and an in vivo analysis of the phenotype of KI mice carrying impaired alpha2-CB interaction. The experimental evidence and the statistical methods employed supports the conclusion drawn.

Major points

1) The Authors observe increased density of small size alpha2-clusters paralleled by a marked decrease of CB clusters. They deduce that such small alpha2 clusters could be extrasynaptic thus indicating a loss of synaptic localization for alpha2-containing receptors. Although this scenario is reasonable, two or three-color immunostaining experiments showing alpha2 co-localization with VGAT and/or CB would provide a more direct evidence of the alpha2-GABAARs altered synaptic clustering in Gabra2-1 mice. As I understand that this experiment could be challenging in acute slices, it could be even performed as a proof

of principle in primary cultures. These results would nicely corroborate the already presented data on alpha2 synaptic/extrasynaptic localization.

2) In a related point, is the loss of alpha2-containing GABAARs at AIS synapses compensated by the inclusion of other GABAARs subtypes?

3) The $\alpha 2$ selective benzodiazepine AZD7325 is reported to ameliorate anxiety and to reduce alterations in the electroencephalogram gamma-power in Gabra2-1 mice. However, an effort could be done to better identify the mechanisms of such effects. The Authors show that the total amount of alpha2 GABAARs in Gabra2-1 mice is increased while they should be less expressed at synapses. Do the Authors think that the beneficial effects of AZD7325 are due to the enhancement of the synaptic currents mediated by the remaining synaptic alpha2-GABAAR receptors? May the AZD7325-mediated effects also arise from increased tonic current mediated by the large amount of extrasynaptic alpha2 GABAARs? A straightforward experiment could be to record sIPSCs from Gabra2-1 mice and see whether AZD7325 can (at least partially) rescue the normal sIPSCs kinetics. This would reinforce the hypothesis the faster kinetics and/or smaller amplitude of inhibitory synaptic currents underlie altered EEG and behavior through disinhibition of the AIS. As a side note, by dissecting out the contribution of sIPSCs amplitude or kinetics one could hypothetically draw important conclusions about the integration window for the AP gating in principal cells.

4) As the Authors mention, alpha2-containing receptors are located not only in the AIS but also, for instance in the hippocampus at somatic synapses between CCK-positive interneurons and pyramidal cells. Interestingly, in the present work it is reported that, upon impairment of alpha2-CB interaction, only AIS synapses (likely formed by chandelier cells expressing PV) show a selective loss of alpha2. Which is the proposed mechanism for such specificity? Is there any other synaptic protein matching the same spatial segregation? For instance, could the DGC (dystrophin glycoprotein complex) which is present at the somatic somatic synapses but not at the AIS synapses (Panzanelli et al., 2011) play a role in such selectivity? The reasons of the selective alpha2 loss in the AIS should be discussed.

5) At page 12 the Authors state: "Gabra2-1 mice have compromised inhibitory synaptic transmission that may be related to decreased $\alpha 2$ cluster size". While I understand how smaller synaptic cluster may lead to reduced sIPSCs amplitude it is less straightforward to understand how they should lead to faster sIPSCs kinetics. Indeed sIPSCs kinetics is largely determined by GABAARs gating properties whereas it does not depend on receptor number. This sentence should be clarified. Which is the reason for sIPSCs faster decay time? As mentioned in point 2, may alpha1-containing receptors replacing alpha2-1-containing receptors leading to sIPSCs speed up? Do the Authors expect that the kinetics of alpha2-1 containing receptors is similar to that of the alpha2-containing ones? The source of sIPSCs should be identified or at least hypothesized.

Minor point:

1) The expressions "prompted by..." or "prompted us..." are probably used too often. I suggest varying these locutions through the MS.

Reviewers' comments:

Reviewer #1 (Remarks to the Author):

This paper describes a novel and unique high affinity interaction between the GTP exchange factor collybistin and a cytoplasmic loop peptide of the $\alpha 2$ subunit of GABAARs, along with a putative role of this interaction in the synaptic accumulation of these receptors in the postsynaptic membrane. The authors had previously described this same $\alpha 2$ sequence as the site of interaction with the subsynaptic scaffolding protein, gephyrin. Others had shown evidence for a trimeric complex between the $\alpha 2$ subunit, gephyrin and collybistin in yeast-two-hybrid assays. In this paper here, the authors used detailed analyses of gephyrin- $\alpha 2$ and collybistin- $\alpha 2$ interactions with purified proteins in vitro to show that gephyrin and collybistin compete for interaction with the $\alpha 2$ subunit, rather than forming a trimeric complex. Mutation of the gephyrin/collybistin binding site of the $\alpha 2$ subunit by gene targeting in mice provides evidence that this interaction is critically important for accumulation of $\alpha 2$ receptors at synapses and also for stability and synaptic localization of collybistin. Behaviorally the mutant mice show increased mortality and spontaneous behavioral and EEG seizures, along with an anxiety phenotype that is reversible with the $\alpha 2$ specific benzodiazepine, AZD7325, but not with the broadly active benzodiazepine, diazepam. This manuscript for the first time provides a plausible mechanism for the differential synapse-specific accumulation of different subtypes of postsynaptic GABAARs that differ in their alpha subunit. The study is overall very well-conceived and comprehensive, and the experiments satisfy a very high standard. It should be of interest to the broad readership of Nature Communications. My concerns are relatively minor and they should be readily addressable.

Main concerns:

The observed loss of CB staining in Gabra2-1 neurons is surprising and unexplained. To differentiate between loss of CB accumulation due to loss of interaction with receptors and due to degradation of CB it would be important to test whether loss of CB is evident by western blot and whether this phenotype is replicated in $\alpha 2$ KO mice.

To address the referee's concern, we have performed additional experimentation. Using immunoblotting we demonstrate that total CB expression is significantly reduced in forebrain lysate from Gabra2-1 heterozygous and homozygous mice, while total gephyrin expression remains consistent across genotypes. It is interesting to speculate on the mechanism linking the Gabra2-1 mutation to CB steady state expression, which may include changes in transcription, translation, or protein half-life. However, we believe that directly investigating this phenomenon, which is technically challenging in brain tissue, lies outside the boundary of the present studies.

To date nothing has been published on the expression of CB in the GABA_A receptor $\alpha 2$ KO mouse, and we were not able to get information (or tissue) through inquiries with collaborators working in the GABA_AR field. They are also not available from known repositories such as Jackson Labs. It is known that CB-KO mice show a loss of $\alpha 2$ containing GABA_A receptors in CA1 pyramidal cells. This loss in pyramidal cells is accompanied by a loss of gephyrin, and $\alpha 1$ containing GABA_A receptors, while PV⁺ interneurons (and thalamic VB neurons) retain gephyrin and $\alpha 1$ -GABA_A receptor expression upon CB KO (Papadopoulos T, Eulenburg V, Reddy-Ailla S, Mansuy IM, Li Y, Betz H (2008) Collybistin is required for both the formation and maintenance of GABAergic postsynapses in the hippocampus. Mol Cell Neurosci 39:161–169). This may suggest that CA1 pyramidal cells express predominantly heteromeric $\alpha 1$ - $\alpha 2$ receptors, which rely upon CB and $\alpha 2$ for expression and retention at synapses, while other cells expressing GABA_A receptors with two $\alpha 1$ subunits are not impacted by the loss of CB/ $\alpha 2$ (Molecular and functional heterogeneity of GABAergic synapses. Fritschy JM, Panzanelli P, Tyagarajan SK. Cell Mol Life Sci. 2012 Aug;69(15):2485-99. doi: 10.1007/s00018-012-0926-4. Review.). We now also discuss this in the manuscript.

Fig 1 I,J shows a relatively modest loss of interaction between mutant Gabra2-1 cytoplasmic loop fragment and CB that is difficult to reconcile with virtually complete loss of mutant receptors from AISs in neurons. It would

seem important to know whether loss of interaction between Gabra2-1 and CB is also evident and perhaps even more dramatic in the context of intact heteromeric ($\alpha 2\beta 2$) receptors.

It is difficult to directly compare observations in neurons with those from expression systems, particularly regarding protein-protein interactions and the formation of inhibitory synapses. It is important to note that the experiments performed in vitro use non-equilibrium conditions, whilst those in cell lines for whole receptors are performed at steady state, under saturating conditions, which will be driven in part by low affinity interactions. While it would be of interest to measure the interaction between CB and whole GABA_A receptors to reveal differences in affinity between these 2 proteins, such experiments are confounded as they are performed under equilibrium conditions. They are also technically challenging as they require the co-expression of 4 distinct cDNAs at the appropriate levels in expression systems. We have attempted to perform co-immunoprecipitation experiments in Gabra2-1 mice in response to reviewer's comments, but these experiments were unsuccessful. GABA_A receptor co-IP has been a long standing challenge in the field due to antibody limitations. Experiments in the Gabra2-1 mouse are made more difficult as the best antibody is directed at the $\alpha 2$ loop which does not recognize the Gabra2-1 mutant subunit (Figure 2). We have shown in other experiments employing a mouse expressing a tagged form of $\alpha 2$, that $\alpha 2$ and CB co-IP from brain lysate (Proteomic Characterization of Inhibitory Synapses Using a Novel pHluorin-tagged γ -Aminobutyric Acid Receptor, Type A (GABAA), $\alpha 2$ Subunit Knock-in Mouse. Nakamura Y, Morrow DH, Modgil A, Huyghe D, Deeb TZ, Lumb MJ, Davies PA, Moss SJ. J Biol Chem. 2016 Jun 3;291(23):12394-407. doi: 10.1074/jbc.M116.724443. Epub 2016 Apr 4.), but again these studies have not been possible with untagged versions of the receptor subunit. Finally, studies in yeast using reporter assays strongly suggest that residues 360-375 in the $\alpha 2$ subunit are important determinants for selective CB binding to this receptor subunit. This manuscript has been cited in our revised paper.

As indicated by the authors in the Discussion, unaltered VGAT staining along pyramidal cell bodies of Gabra2-1 mice cannot be taken as evidence that somatic synapses are unaffected, and indeed this is one significant aspect of this manuscript that is unresolved. Given that loss of CB and $\alpha 2$ immunoreactivity is not limited to AIS synapses it seems likely that defects in (or altered) innervation are not limited to AIS synapses either. If VGAT+ synapses are unchanged and PV+ synapses are reduced (as evidenced at AISs), then one might predict a compensatory increase in somatic synapses from CCK basket cells, which the authors could quantitate by means of postsynaptic dystroglycan clusters.

We have performed additional experimentation to stain for dystroglycan, however two different antibodies failed to produce satisfactory staining. We followed up with Western blot analysis of dystroglycan, which did not reveal any significant difference between wildtype and Gabra2-1 heterozygous and homozygous mice.

Fig 7C, D. To ascertain the anxiety phenotype and anxiolytic-like effect of AZD7325, the authors should report % open arm entries and # of total arm entries or distance traveled. Please report all parameters for both genotypes.

We have added this information in the revised manuscript, and the graphs of open arm entries, and total distance traveled are now shown in Supplemental Figure 5. Briefly, we found that analysis of the number of open arm entries supported the anxiety phenotype indicated by assessment of percent open arm time, with Gabra2-1 mice showing fewer open arm entries compared to wildtype littermate controls. This index of anxiety was ameliorated by AZD7325 treatment of Gabra2-1 mice, but not by diazepam, again similar to the assessment of % open arm time. Using entries as an index of anxiety did not yield a significant difference between AZD7325 and vehicle in wildtype mice ($p=0.068$). Assessment of total distance traveled in the elevated plus maze did not reveal a significant difference between genotypes, and diazepam and AZD7325 treatment did not significantly differ from vehicle control treatment in terms of total distance traveled by mice in these treatment groups.

The selective anxiolytic-like effect of AZD7325 vs diazepam of het mutants in the EPM is interesting but

unexplained. Why would a reduction in AZD7325 targets result in increased behavioral response to AZD7325 but not diazepam – this should be discussed.

To support this finding we conducted EEG analysis on the effects of diazepam in WT and Gabra2-1 heterozygotes. We have added data that demonstrates that diazepam reduces δ -power in wildtype mice as has been previously reported in the literature, but does not do so in Gabra2-1 mice, supporting the behavior data. This new data is shown in supplemental Figure 8. Previous studies have suggested that binding of benzodiazepines to $\alpha 2$ containing receptors is required for the suppression of EEG δ power (Kopp, C., Rudolph, U., Löw, K. & Tobler, I. Modulation of rhythmic brain activity by diazepam: GABA(A) receptor subtype and state specificity. *Proc. Natl. Acad. Sci. U. S. A.* 101, 3674–3679. 2004), while $\alpha 1$ and $\alpha 3$ binding has a limited role in δ suppression (Kopp, C., Rudolph, U., Keist, R. & Tobler, I. Diazepam-induced changes on sleep and the EEG spectrum in mice: role of the alpha3-GABA(A) receptor subtype. *Eur. J. Neurosci.* 17, 2226–2230. 2003; Tobler, I., Kopp, C., Deboer, T. & Rudolph, U. Diazepam-induced changes in sleep: role of the alpha 1 GABA(A) receptor subtype. *Proc. Natl. Acad. Sci. U. S. A.* 98, 6464–6469. 2001). This difference in effect of diazepam in WT vs Gabra2-1 heterozygous mice may be related to the predominance of $\alpha 1$ - and $\alpha 3$ - mediated effects of diazepam when $\alpha 2$ containing receptors are compromised as in Gabra2-1 mice. In contrast, since AZD7325 acts preferentially on $\alpha 2$ containing receptors, it remains effective at suppressing δ and reducing anxiety-like behavior. We now discuss this result in our revised manuscript.

Please add information on statistics including n, F and p values, and the type of tests used, to the text or figure legends for all experiments throughout the manuscript.

This information has been added to our revised manuscript in the figure legends (mean, SE, p values), methods (n's) and supplemental material (table of statistics, mean, SE, F, and p values).

Other Details:

The text describing Fig 1C says that “a3L did not alter the elution profile of the SH3 domain”. To my eyes the elution profile of SHR in the presence of a3L looks the same as that in the presence of a1L. What am I missing? And how was the affinity of the a2L-SH3 complex estimated to be in the micro molar range, as part of Fig 1C?

The reviewer is correct in stating that there is essentially no difference between the elution profiles of the $\alpha 3L$ and $\alpha 1L$ constructs and we have reworded the text accordingly, apologies for this error. The estimation of the of the $\alpha 2L$ -SH3 complex affinity in the micromolar range is based on the protein concentrations employed in the experiment, and affinity was determined through experiments shown in Figure 1D.

In Suppl Figure 1B it would be helpful to add the Gabra1 sequence for comparison.

We have added the sequence of the GABA_A receptor $\alpha 1$ subunit aligned with that for the $\alpha 2$ subunit to Suppl Figure 1 (panel D), as suggested.

Legend of Fig 1 reads “... complete saturation of GephE with the GABAAR $\alpha 2$ subunit upon incubation with an 8-fold excess of the receptor loop.” Where does this 8-fold ratio come from?

The wording of the figure legend for Figure 1 has been modified to make it more clear and succinct, thank you for calling attention to the confusing wording.

What exactly does “expressed CB” refer to? Do you mean CB-containing crude extracts?

We have modified the text describing these experiments to provide more clear explanation of the assay and results.

The text on page 9 referring to Fig 2B-D and the corresponding Figure Legend need to be carefully revised with reference to the proper parts of this Figure.

The figure has been modified to include new data, and we have checked the figure legend to ensure that it appropriately matches the figure, apologies for the labeling error in the prior version.

The text indicates that $\alpha 2$ expression in basal forebrain and hippocampus of Gabra2-1 mice is increased. Please include this data in Fig 2G. The text indicates that expression was increased in the molecular layer but this info is missing from Fig 2G. Presumably, the data labeled CA1 should be labeled pyramidal layer?

The figure has been modified, and the full analysis of DAB immunoreactivity across brain regions is now included in supplemental figure 2, along with representative images. We have modified the labeling to be more consistent and clear, thank you for the suggestion.

What were the genotypes of the mice quantified in Fig 6A-C? Can the data be reported separately for heterozygous and homozygous mutant offspring? Fig 6D, E are unclear also – it looks like the genotypes denoted along the X-axis refer to the parental mice but this is not stated in the Figure legend. Again, this data would be more informative if the data were disaggregated by genotype. Similarly, in Fig 6G it would be interesting to know whether heterozygotes were different from homozygotes.

We have conducted additional analysis of genotype on tissue from dead pups that were available. Overall this analysis revealed approximately 2 fold more heterozygous offspring dead compared to homozygous offspring, however the majority of our matings were het x het crosses that would be expected to produce a 1:2:1 ratio, meaning that likely 2 fold greater proportions of hets are being produced as compared to homos. We have exchanged panel 6B with a graph that now shows the genotype information available for pups found dead from het x het crosses, corrected for the proportion produced based on Mendelian ratios. This analysis reveals that homozygous mice are more likely to die at earlier ages compared to heterozygous mice, but that both are susceptible to early mortality.

In figure 6G, repeated measures ANOVA did not produce a significant difference between heterozygous and homozygous mice.

Data in Fig 7 F are difficult to read. I suggest using a color code for genotype.

We have changed the color of the graph so that the data are more clearly distinguishable. We have applied this color scheme through the EEG FFT panels in Fig 7 (panel 7G), and in the supplemental data (Supplemental Figure 8).

Reviewer #2 (Remarks to the Author):

This manuscript reports an interaction of the GEF collybistin (CB) with the intracellular loop of the GABAAR $\alpha 2$ subunit and synaptic, electrophysiological and behavioural changes in line with decreased GABAergic inhibition observed upon replacing the putative CB binding site in the mouse $\alpha 2$ gene by the respective GABAAR $\alpha 1$ sequence. As the mechanism of CB mediated gephyrin and GABAAR clustering is still enigmatic, this is basically an important result. However, after looking at the data provided I have considerable doubts about the authors' conclusions.

First, the binding data presented in Fig. 1 and Suppl. Table 2 are not in agreement with each other. The NAGE results showing comparable binding of CB's SH3 domain to both $\alpha 1L$ and $\alpha 2L$ are difficult to reconcile with a >500-fold difference in K_d determined by ITC. Also, the $\alpha 3L$ peptide appears not an adequate control, as it is considerably shorter and not as hydrophobic as the $\alpha 1L$ and in particular the $\alpha 2L$ sequences. Furthermore, all these quantitative interaction experiments are performed with the isolated SH3 domain and not recombinant

CB. Second, no evidence for specificity of CB binding to the full-length $\alpha 2$ subunit is provided. Co-immunoprecipitation experiments as shown in Fig. 1I for $\alpha 2$ and $\alpha 2-1$ should also be performed for the full-length $\alpha 1$ and $\alpha 3$ subunits and ideally be complemented with immunoprecipitations from brain extracts in order to corroborate that CB binding is indeed $\alpha 2$ specific and occurring in vivo

The NAGE experiments served as an initial technique to compare the relative affinities of the three receptor loops which were subsequently refined by analytical size exclusion chromatography and isothermal titration calorimetry. The salient point of the NAGE data presented is that $\alpha 2$ retards the migration of CB compared to $\alpha 3$ where no shift is seen. With $\alpha 1$ a smaller shift is seen. To clarify this issue, we have an additional red arrow indicating the shift in CB migration seen with $\alpha 2$.

At present the only viable means of definitively measuring the binding affinity of portions of GABA_ARs with other proteins is via methods such as ITC. The issue of core CB binding motif hydrophobicity is not of significance as we perform our measurements in with soluble fusion proteins encoding the entire intracellular domain of the $\alpha 1$ or $\alpha 2$ subunits. We fully agree that it would be of interest to perform these studies with full-length collybistin, however previous studies have shown that it is not possible to obtain sufficient quantities of this protein. We disagree with the reviewer's assessment that we did not provide any evidence for an $\alpha 2$ -specific binding of CB as our combination of NAGE, Gel filtration and ITC provides data showing an approximate 500 fold affinity of CB for this receptor subunit compared to $\alpha 1$. ITC is the gold standard measurement for protein affinity and the use of this approach in our studies clearly defines a high affinity binding of CB to the $\alpha 2$ subunit. Importantly, this result is consistent with published studies in yeast showing selective binding of $\alpha 2$ to CB, cited in our revised paper.

We did attempt to perform co-immunoprecipitation experiments in Gabra2-1 mice in response to reviewer's comments, but these experiments were unsuccessful. GABA_A receptor co-IP has been a long standing challenge in the field due to antibody limitations. Consistent with this, a number of published studies have failed to demonstrate co-immunoprecipitation of GABA_ARs and CB. Experiments in the Gabra2-1 mouse are made more difficult as the best antibody is directed at the $\alpha 2$ loop which does not recognize the Gabra2-1 mutant subunit (Figure 2). We have also shown in previously published experiments employing a mouse expressing a tagged form of $\alpha 2$, that $\alpha 2$ and CB co-IP from brain lysate (Proteomic Characterization of Inhibitory Synapses Using a Novel pHluorin-tagged γ -Aminobutyric Acid Receptor, Type A (GABAA), $\alpha 2$ Subunit Knock-in Mouse. Nakamura Y, Morrow DH, Modgil A, Huyghe D, Deeb TZ, Lumb MJ, Davies PA, Moss SJ. J Biol Chem. 2016 Jun 3;291(23):12394-407. doi: 10.1074/jbc.M116.724443. Epub 2016 Apr 4.), but again these studies have not been possible with untagged versions of the receptor subunit.

Another major concern relates to the central $\alpha 1$ sequence substituted into the Gabra2-1 mice. In ITC, the corresponding $\alpha 2$ peptide ($\alpha 2$ min2) shows no significant binding to the SH3 domain, and the difference in CB intensity co-precipitated with the purified $\alpha 2$ and $\alpha 2-1$ subunits (Fig. 1I) is only small (about 30%; see Fig. 1J). Hence one wonders why an exchange of this motif was chosen for generating the Gabra2-1 mice; I honestly have difficulties to understand the authors' strategy here. One argument they mention is conformation dependence of CB binding. However, I have not been able to detect evidence in support of this idea; rather, the fact that binding experiments can be performed with soluble tagged loop peptides seems contradictory.

Our ITC experiments suggest that residues 360-375 within the ICD of the $\alpha 2$ subunit mediate high affinity binding to CB, 500 fold compared to $\alpha 1$. Using deletion analysis we then reveal that this preferential binding is mediated via amino acid residues 360-375 in the $\alpha 2$ subunit. A general difference in the binding strength between the core binding region of a receptor and the full-length cytoplasmic loop when interacting with a partner protein is not surprising. In the case of gephyrin ITC binding assays with short peptides derived from the receptors are only possible for the glycine receptor β subunit and to a limited degree with the GABA_A receptor $\alpha 3$ subunit. Consequently we were not surprised that we had to use the full-length intracellular loops of the GABA_AR $\alpha 1/2/3$ subunits in our studies and it is hence not surprising that the $\alpha 2$ min2 peptide does not show measurable binding (Table S2). Nevertheless this core region must be the determinant of binding since this is where the

three α subunits differ in primary sequence. Thus to test the role that CB binding plays in regulating the formation of inhibitory synapses containing $\alpha 2$ subunit we made a mouse in which these core residues were substitute to those with $\alpha 1$. Critically, we show that this mutation reduces CB binding to full-length $\alpha 2$ subunits under steady state, saturating conditions when co-expressed in mammalian cells (Fig X).

Regarding “conformation” we again apologize for our lack of clarity; this term is used in reference to the novel nature of the CB- $\alpha 2$ interaction. Most SH3 domains bind to proline rich sequences in target proteins, however the binding motif for CB does not contain prolines thus we suggest that the novel motif with $\alpha 2$ may have a unique conformation/structure compared to the established canonical proline motifs found in other proteins.

The analysis of the Gabra2-1 mice is overall straightforward and presents many results consistent with reduced synaptic inhibition. The immunocytochemistry presented in Fig. 3 and Suppl. Fig. 2, however, poses a number of questions. As noted by the authors themselves, in Gabra2-1 sections $\alpha 2$ staining is much more diffuse than in WT specimens, suggesting (but not showing) that a major fraction of the $\alpha 2$ receptors is not synaptically localized. This should be evaluated by providing colocalization, and not just density, values for $\alpha 2$ and VGAT immunoreactivities, as done in Fig. 5. Ideally, colocalization values should also be determined for $\alpha 2$ and gephyrin both in sections and the AIS, since gephyrin constitutes a major binding partner of CB and has been shown to be required for $\alpha 2$ synaptic localization in vivo. Hence the loss of CB clusters could be a consequence of reduced gephyrin clustering at synaptic sites. The authors' proposal that CB might be eventually degraded in the Gabra2-1 mice could be substantiated by immunoblots of WT and mutant brain extracts.

In response to your comment we have performed experiments to examine the co-localization of $\alpha 2$ and VGAT. These experiments are now presented as Supplemental Figure 4. Briefly, these experiments show that the percentage of VGAT clusters colocalized with $\alpha 2$ does not change between wildtype and Gabra2-1 mice. We were able to detect a difference in the percentage of $\alpha 2$ that is colocalized with VGAT, presumably due to the more diffuse or extrasynaptic localization of receptors containing the mutant subunit.

Immunohistochemistry for $\alpha 2$ combined with gephyrin for colocalization analysis failed to produce satisfactory staining. We followed up with Western blotting experiments to examine total collybistin and total gephyrin expression, now shown in Figure 2. In keeping with our immunofluorescence data there is a significant reduction in the level of CB expression in forebrain lysates from Gabra2-1 mice. Total expression of gephyrin in contrast is not changed in Gabra2-1 forebrain lysate compared to wildtype controls. Beyond the support that the new data provides, it is also important to note that the total number of synapses VGAT clusters, and the number of VGAT clusters on the soma do not change in the Gabra2-1 mice. To add further support for this finding, we now also include analysis of GAD-67 staining which shows no difference between WT and Gabra2-1 mice.

Another point that should be considered is the strong overexpression of the $\alpha 2-1$ subunit seen in at least some of the Gabra2-1 mice (Fig. 2E, F). This could result in dominant-negative effects which might contribute to the electrophysiological and behavioural deficits described. For this reason, it also appears mandatory to analyze a sufficient number of animals electrophysiologically. The mean values presented in Suppl. Fig. 3B are derived from only 2-3 animals. In my opinion, this is not sufficient for a reliable statistical analysis. Also, it appears desirable that precise animal n numbers are given for the quantitative results presented in several of the main figures.

We now include analysis from 5-6 mice per genotype. We also now more consistently report the n numbers (methods section), as well as means, standard errors, and more complete p values (figure captions, supplemental material) for our experiments throughout the manuscript, thank you for calling our attention to this oversight.

In summary, although this study contains a plethora of significant data, I remain to be convinced that the statement made in the title of this manuscript is indeed correct. I therefore do not comment in detail on the

other data presented on the survival, behaviour and pharmacological rescue of the Gabra2-1 mice, which are indicative of deficient GABAAR $\alpha 2$ neurotransmission.

Reviewer #3 (Remarks to the Author):

The manuscript "Developmental seizures and mortality result from reducing the affinity of the GABAA Receptor $\alpha 2$ subunit for collybistin" by Hines et al., investigates the molecular mechanisms of the GABAAR subtype localization in specific synapse subsets. In particular the Authors identify a high affinity binding between the scaffold protein collybistin (CB) and the $\alpha 2$ subunit of GABAA receptors that is responsible for the localization of $\alpha 2$ -containing GABAA receptors in the axon initial segment, a sub-cellular compartment that is crucial for the control of neuronal excitability. In line with this, the Authors demonstrate that KI mice bearing a chimeric $\alpha 2$ subunit with disrupted CB binding show increased susceptibility to seizures, anxiety and early mortality.

The findings of the present work are very relevant. Indeed, this manuscript is shedding light on the mechanisms responsible for the clustering of GABAA receptor subtypes at specific synapses and/or sub-neuronal compartments, a subject that remains poorly understood. In addition the functional implications of the $\alpha 2$ -CB interaction are shown at behavioral level thus linking malfunction of the synaptic function at the molecular level to the possible etiology of severe neurological disease.

This study is well conducted and employs a state-of-the-art multi-level approach that encompasses sophisticated biochemical techniques, electrophysiology and an in vivo analysis of the phenotype of KI mice carrying impaired $\alpha 2$ -CB interaction. The experimental evidence and the statistical methods employed supports the conclusion drawn.

Major points

1) The Authors observe increased density of small size $\alpha 2$ -clusters paralleled by a marked decrease of CB clusters. They deduce that such small $\alpha 2$ clusters could be extrasynaptic thus indicating a loss of synaptic localization for $\alpha 2$ -containing receptors. Although this scenario is reasonable, two or three-color immunostaining experiments showing $\alpha 2$ co-localization with VGAT and/or CB would provide a more direct evidence of the $\alpha 2$ -GABAARs altered synaptic clustering in Gabra2-1 mice. As I understand that this experiment could be challenging in acute slices, it could be even performed as a proof of principle in primary cultures. These results would nicely corroborate the already presented data on $\alpha 2$ synaptic/extrasynaptic localization.

In response to your comment we have performed experiments to examine the co-localization of $\alpha 2$ and VGAT. These experiments are now presented as Supplemental Figure 4. Briefly, these experiments show that the percentage of VGAT clusters colocalized with $\alpha 2$ does not change between wildtype and Gabra2-1 mice. We were able to detect a difference in the percentage of $\alpha 2$ that is colocalized with VGAT, presumably due to the more diffuse or extrasynaptic localization of receptors containing the mutant subunit.

We followed up with Western blotting experiments to examine total collybistin expression, now shown in Figure 2. In keeping with our immunofluorescence data there is a significant reduction in the level of CB expression in forebrain lysates from Gabra2-1 mice.

2) In a related point, is the loss of $\alpha 2$ -containing GABAARs at AIS synapses compensated by the inclusion of other GABAARs subtypes?

We believe the replacement of $\alpha 2$ with other subunits is unlikely as the total number of VGAT puncta on the AIS are reduced in cortical sections from Gabra2-1 mice. This has been more clearly described and stated in our revised manuscript.

3) The $\alpha 2$ selective benzodiazepine AZD7325 is reported to ameliorate anxiety and to reduce alterations in the

electroencephalogram gamma-power in Gabra2-1 mice. However, an effort could be done to better identify the mechanisms of such effects. The Authors show that the total amount of alpha2 GABAARs in Gabra2-1 mice is increased while they should be less expressed at synapses. Do the Authors think that the beneficial effects of AZD7325 are due to the enhancement of the synaptic currents mediated by the remaining synaptic alpha2-GABAAR receptors? May the AZD7325-mediated effects also arise from increased tonic current mediated by the large amount of extrasynaptic alpha2 GABAARs? A straightforward experiment could be to record sIPSCs from Gabra2-1 mice and see whether AZD7325 can (at least partially) rescue the normal sIPSCs kinetics. This would reinforce the hypothesis the faster kinetics and/or smaller amplitude of inhibitory synaptic currents underlie altered EEG and behavior through disinhibition of the AIS.

As a side note, by dissecting out the contribution of sIPSCs amplitude or kinetics one could hypothetically draw important conclusions about the integration window for the AP gating in principal cells.

In our manuscript we show that Gabra2-1 mice have enhanced EEG δ power, a phenomenon that is widely believed to correlate with increased anxiety, which is consistent with the elevated anxiety-like behavior of the mice. We show that AZD7325, but not diazepam, rescues the deficits in EEG activity and anxiety in the mutant mice. We have added data that demonstrates that diazepam reduces δ -power in wildtype mice as has been previously reported in the literature, but does not do so in Gabra2-1 mice, supporting the behavior data. This new data is shown in supplemental Figure 8. Previous studies have suggested that binding of benzodiazepines to α 2 containing receptors is required for the suppression of EEG δ power (Kopp, C., Rudolph, U., Löw, K. & Tobler, I. Modulation of rhythmic brain activity by diazepam: GABA(A) receptor subtype and state specificity. *Proc. Natl. Acad. Sci. U. S. A.* 101, 3674–3679. 2004), while α 1 and α 3 binding has a limited role in δ suppression (Kopp, C., Rudolph, U., Keist, R. & Tobler, I. Diazepam-induced changes on sleep and the EEG spectrum in mice: role of the alpha3-GABA(A) receptor subtype. *Eur. J. Neurosci.* 17, 2226–2230. 2003; Tobler, I., Kopp, C., Deboer, T. & Rudolph, U. Diazepam-induced changes in sleep: role of the alpha 1 GABA(A) receptor subtype. *Proc. Natl. Acad. Sci. U. S. A.* 98, 6464–6469. 2001). This difference in effect of diazepam in WT vs Gabra2-1 heterozygous mice may be related to the predominance of α 1- and α 3- mediated effects of diazepam when α 2 containing receptors are compromised as in Gabra2-1 mice. In contrast, since AZD7325 acts preferentially on α 2 containing receptors, it remains effective at suppressing δ and reducing anxiety-like behavior. We now discuss this result in our revised manuscript. These experiments suggest that the deficits in α 2 subunit expression in the mutant mice are linked to the deficits in EEG power and behavior seen in the Gabra2-1 mouse. We agree with the referee that further defining the cellular mechanism by which ADZ7325 exerts this selective anxiolytic action is of significance. However, we believe that to fully address this complex issue warrants a separate, dedicated study.

4) As the Authors mention, alpha2-containing receptors are located not only in the AIS but also, for instance in the hippocampus at somatic synapses between CCK-positive interneurons and pyramidal cells. Interestingly, in the present work it is reported that, upon impairment of alpha2-CB interaction, only AIS synapses (likely formed by chandelier cells expressing PV) show a selective loss of alpha2. Which is the proposed mechanism for such specificity? Is there any other synaptic protein matching the same spatial segregation? For instance, could the DGC (dystrophin glycoprotein complex) which is present at the somatic somatic synapses but not at the AIS synapses (Panzanelli et al., 2011) play a role in such selectivity? The reasons of the selective alpha2 loss in the AIS should be discussed.

We have performed additional experimentation to stain for dystroglycan, however two different antibodies failed to produce satisfactory staining. We followed up with Western blot analysis of dystroglycan, which did not reveal any significant difference between wildtype and Gabra2-1 heterozygous and homozygous mice.

The specific impact of Gabra2-1 mutation on AIS inhibitory synapses may also be related to interaction of the α 2-CB complex with other proteins that have been implicated in the stabilization of AIS functional components such as ankyrin_E (Kordeli, E., Lambert, S. & Bennett, V. Ankyrin A NEW ANKYRIN GENE WITH NEURAL-SPECIFIC ISOFORMS LOCALIZED AT THE AXONAL INITIAL SEGMENT AND NODE OF RANVIER. *J. Biol. Chem.* 270, 2352–2359. 1995), neurofascin (Kriebel, M. et al. The Cell Adhesion Molecule Neurofascin Stabilizes Axo-axonic GABAergic Terminals at the Axon Initial

Segment. *J. Biol. Chem.* 286, 24385–24393. 2011), NrCAM (Jenkins, S. M. & Bennett, V. Ankyrin-G coordinates assembly of the spectrin-based membrane skeleton, voltage-gated sodium channels, and L1 CAMs at Purkinje neuron initial segments. *J Cell Biol* 155, 739–746. 2001), and β IV-spectrin (Berghs, S. *et al.* betaIV spectrin, a new spectrin localized at axon initial segments and nodes of ranvier in the central and peripheral nervous system. *J. Cell Biol.* 151, 985–1002. 2000). Neurofascin in particular has been shown to impact the clustering of GABA_ARs and GAD positive terminals on the AIS making it an interesting target for further study. We now discuss these potential mechanisms for selective loss at the AIS synapses in the revised manuscript.

5) At page 12 the Authors state: “Gabra2-1 mice have compromised inhibitory synaptic transmission that may be related to decreased α 2 cluster size”. While I understand how smaller synaptic cluster may lead to reduced sIPSCs amplitude it is less straightforward to understand how they should lead to faster sIPSCs kinetics. Indeed sIPSCs kinetics is largely determined by GABA_ARs gating properties whereas it does not depend on receptor number. This sentence should be clarified. Which is the reason for sIPSCs faster decay time? As mentioned in point 2, may alpha1-containing receptors replacing alpha2-1-containing receptors leading to sIPSCs speed up? Do the Authors expect that the kinetics of alpha2-1 containing receptors is similar to that of the alpha2-containing ones? The source of sIPSCs should be identified or at least hypothesized.

Previous studies suggest that IPSC decay is in part determined by GABA_AR α subunit isoforms. This enhanced decay would be consistent with a loss of α 2 subunit containing GABA_ARs, and their replacement with those containing α 1 subunits. This has been discussed in our revised manuscript, and relevant papers cited (Eyre, M. D., Renzi, M., Farrant, M. & Nusser, Z. Setting the time course of inhibitory synaptic currents by mixing multiple GABA(A) receptor α subunit isoforms. *J. Neurosci. Off. J. Soc. Neurosci.* 32, 5853–5867. 2012; Goldstein, P. A. *et al.* Prolongation of hippocampal miniature inhibitory postsynaptic currents in mice lacking the GABA(A) receptor alpha1 subunit. *J. Neurophysiol.* 88, 3208–3217. 2002).

Minor point:

1) The expressions “prompted by...” or “prompted us...” are probably used too often. I suggest varying these locutions through the MS.

Thank you for altering us to the repetitive used of this expression which has been avoided in the revised version.

Reviewers' comments:

Reviewer #1 (Remarks to the Author):

The authors have adequately responded to my concerns, including the incorporation of additional data where feasible, and corrections of the text and additional discussion of unexplained findings. I have no further comments

Reviewer #2 (Remarks to the Author):

I am pleased with the authors' revisions in response to my concerns about the low number of animals used for the electrophysiological recordings and the lack of colocalization values for $\alpha 2$ and VGAT immunoreactivities, although the density of VGAT sites detectable in the new Suppl. Fig. 4 appears low. This may relate to the specific region shown/magnification used; the figure legend does not specify the area presented, nor is the length of the size bar indicated.

The authors' responses to my major concern, namely that their data do not prove that the phenotype of the Gabra2-1 mice is due to the loss of CB binding to the $\alpha 2$ subunit, are in my view, however, unsatisfactory. First, data on gephyrin density at $\alpha 2$ -positive synapses have not been obtained; thus, it cannot be excluded that the lower $\alpha 2$ content of GABAergic postsynapses is due to a disturbed $\alpha 2$ -gephyrin interaction resulting from the substitution of the gephyrin/CB binding domain in the $\alpha 2L$ loop. The fact that, in contrast to CB, gephyrin total levels as determined by Western blotting are not changed in Gabra2-1 animals is not a convincing argument, since due to its widespread enzymatic function gephyrin is expressed in most, if not all, cells of the body, including astrocytes. Moreover, as the authors admit themselves, experiments showing co-immunoprecipitation of native GABAARs and CB have not been successful, nor has an attempt been made to demonstrate $\alpha 2L$ -specific binding to full-length CB, a point that I consider crucial in view of the demonstrated importance of ligand-SH3 domain interactions for CB conformation and activity. The rebuttal statement that "previous studies have shown that it is not possible to obtain sufficient quantities of this protein" is not correct: one of the authors of this manuscript, Dr. Schindelin, is senior author/co-author of two high-profile studies, in which substantial amounts of recombinant full-length CB-SH3 protein were generated for extensive crystallographic studies (Xiang et al., 2006; and Ref. 41: Soykan et al., 2014).

As further evidence for a selective binding of $\alpha 2$ to CB, the authors now cite their previous proteomic work with a mouse expressing pH-fluorin tagged $\alpha 2$ (Nakamura et al., 2016). In this study, they indeed show that the tagged $\alpha 2$ subunit and CB can be co-isolated from brain lysates. However, in addition 148 (!) other proteins, including not only known GABAAR/CB interaction partners such as gephyrin and neuroligins but also channel, receptor and transporter proteins, protein kinases, GTPases, trafficking proteins etc., were identified in the GABAAR complexes isolated. Thus, the authors' own work vastly extends the number of potential protein- $\alpha 2L$ interactions that may be disturbed in, and cause the phenotype of, the Gabra2-1 mice.

In conclusion, I agree with the authors' statement that their data provide evidence for an in vitro interaction between a2L and the SH3 domain of CB, although in my view there are substantial discrepancies between the NAGE and SEC/ITC results. This, however, is not my major problem with this study - I just remain to be convinced of the conclusion given in the title of the manuscript.

Reviewer #3 (Remarks to the Author):

The Authors satisfactory addressed all my concerns

Reviewers' comments:

Reviewer #1 (Remarks to the Author):

The authors have adequately responded to my concerns, including the incorporation of additional data where feasible, and corrections of the text and additional discussion of unexplained findings. I have no further comments.

Reviewer #2 (Remarks to the Author):

I am pleased with the authors' revisions in response to my concerns about the low number of animals used for the electrophysiological recordings and the lack of colocalization values for $\alpha 2$ and VGAT immunoreactivities, although the density of VGAT sites detectable in the new Suppl. Fig. 4 appears low. This may relate to the specific region shown/magnification used; the figure legend does not specify the area presented, nor is the length of the size bar indicated.

We have revised the $\alpha 2$ VGAT colocalization figure (now Supplemental figure 5) according to your feedback. We have changed the zoom level to make the colocalization more apparent, specified the region shown in the figure caption, and specified the length of the indicated scale bar. Apologies for our oversight on those details.

The authors' responses to my major concern, namely that their data do not prove that the phenotype of the Gabra2-1 mice is due to the loss of CB binding to the $\alpha 2$ subunit, are in my view, however, unsatisfactory. First, data on gephyrin density at $\alpha 2$ -positive synapses have not been obtained; thus, it cannot be excluded that the lower $\alpha 2$ content of GABAergic postsynapses is due to a disturbed $\alpha 2$ -gephyrin interaction resulting from the substitution of the gephyrin/CB binding domain in the $\alpha 2L$ loop. The fact that, in contrast to CB, gephyrin total levels as determined by Western blotting are not changed in Gabra2-1 animals is not a convincing argument, since due to its widespread enzymatic function gephyrin is expressed in most, if not all, cells of the body, including astrocytes.

We now show data from cultured cortical cells demonstrating that, much like the VGAT and $\alpha 2$ colocalization, the percentage of gephyrin clusters that contain $\alpha 2$ is unchanged upon Gabra2-1 mutation. From these images it is also evident that the clustering of gephyrin is not interrupted upon Gabra2-1 mutation. Although we do observe that the percentage of $\alpha 2$ clusters that contain gephyrin is reduced, this is likely due to the increased expression and more diffuse localization of $\alpha 2$, demonstrated in multiple experiments and described in the text.

Moreover, as the authors admit themselves, experiments showing co-immunoprecipitation of native GABAARs and CB have not been successful, nor has an attempt been made to demonstrate $\alpha 2L$ -specific binding to full-length CB, a point that I consider crucial in view of the demonstrated importance of ligand-SH3 domain interactions for CB conformation and activity. The rebuttal statement that "previous studies have shown that it is not possible to obtain sufficient quantities of this protein" is not correct: one of the authors of this manuscript, Dr. Schindelin, is senior author/co-author of two high-profile studies, in which substantial amounts of recombinant full-length CB-SH3 protein were generated for extensive crystallographic studies (Xiang et al., 2006; and Ref. 41: Soykan et al., 2014).

While we agree that it would have been great to perform these experiments also with full-length collybistin, we again must confirm that full-length collybistin is too unstable to purify sufficient quantities and subsequently carry out the binding assays. We disagree with the reviewer's assessment that we did not provide any evidence for an $\alpha 2$ -specific binding of collybistin; our in vitro binding data, in particular the ITC experiments, clearly demonstrate the specific binding. Specifically, the NAGE experiments served as an initial technique to compare the relative affinities of the three receptor loops which were subsequently refined by analytical size exclusion chromatography and isothermal titration calorimetry. Quantitative values of the binding strength can only be derived from the ITC experiments.

As further evidence for a selective binding of $\alpha 2$ to CB, the authors now cite their previous proteomic work with a mouse expressing pH-fluorin tagged $\alpha 2$ (Nakamura et al., 2016). In this study, they indeed show that the tagged $\alpha 2$ subunit and CB can be co-isolated from brain lysates. However, in addition 148 (!) other proteins, including not only known GABAAR/CB interaction partners such as gephyrin and neuroligins but also channel, receptor and transporter proteins, protein kinases, GTPases, trafficking proteins etc., were identified in the GABAAR complexes isolated. Thus, the authors' own work vastly extends the number of potential protein- $\alpha 2$ L interactions that may be disturbed in, and cause the phenotype of, the Gabra2-1 mice.

While we do not argue with the point raised about the large number of other proteins identified in the Nakamura et al., 2016 study, we do disagree with the interpretation of these results as presented by the reviewer. The proteins identified in this study do not implicate direct interaction of each of these identified proteins with the $\alpha 2$ subunit, and certainly do not specify interaction with the large intracellular loop of the $\alpha 2$ subunit, but rather indicate their presence in a large complex associated with GABA_ARs.

In conclusion, I agree with the authors' statement that their data provide evidence for an in vitro interaction between $\alpha 2$ L and the SH3 domain of CB, although in my view there are substantial discrepancies between the NAGE and SEC/ITC results. This, however, is not my major problem with this study - I just remain to be convinced of the conclusion given in the title of the manuscript.

We are open to considering alternative titles for the manuscript if this is a major sticking point.

Reviewer #3 (Remarks to the Author):

The Authors satisfactory addressed all my concerns.

Editorial Note: Following the second round of review, the editor informally consulted with referees #1 and 3 asking them to comment on the review of referee #2. Authors then revised the paper according to the advice recieved in this consultation, resulting in the final round of review in which only referee #3 was further consulted.